# In silico screening by AlphaFold2 program revealed the potential binding partners of nuage-localizing proteins and piRNA-related proteins

**Shinichi Kawaguchi[1]\*[†], Xin Xu[1†], Takashi Soga[2], Kenta Yamaguchi[3], Ryuuya Kawasaki[3], Ryota Shimouchi[4], Susumu Date[2,4], Toshie Kai[1]\***

[1]Graduate School of Frontier Biosciences, Osaka University, Osaka, Japan; [2]D3 Center, Osaka University, Osaka, Japan; [3]NEC Solution Innovators, Ltd., Tokyo, Japan; [4]Graduate School of Information Science and Technology, Osaka University, Osaka, Japan

**\*For correspondence:**
kawaguchi.shinichi.fbs@osaka-u.ac.jp (SK);
kai.toshie.fbs@osaka-u.ac.jp (TK)

[†]These authors contributed equally to this work

## eLife Assessment

This **useful** study employs AlphaFold2 to predict interactions among 20 nuage proteins, identifying 5 novel interaction candidates, 3 of which are validated experimentally through co-immunoprecipitation. Expanding the analysis to 430 oogenesis-related proteins and screening ~12,000 *Drosophila* proteins for interactions with Piwi, the study identifies 164 potential binding partners, demonstrating how computational predictions can streamline experimental validation. This study provides a **solid** basis for further investigations into eukaryotic protein interaction networks.

**Abstract** Protein–protein interactions are fundamental to understanding the molecular functions and regulation of proteins. Despite the availability of extensive databases, many interactions remain uncharacterized due to the labor-intensive nature of experimental validation. In this study, we utilized the AlphaFold2 program to predict interactions among proteins localized in the nuage, a germline-specific non-membrane organelle essential for piRNA biogenesis in *Drosophila*. We screened 20 nuage proteins for 1:1 interactions and predicted dimer structures. Among these, five represented novel interaction candidates. Three pairs, including Spn-E_Squ, were verified by co-immunoprecipitation. Disruption of the salt bridges at the Spn-E_Squ interface confirmed their functional importance, underscoring the predictive model's accuracy. We extended our analysis to include interactions between three representative nuage components—Vas, Squ, and Tej—and approximately 430 oogenesis-related proteins. Co-immunoprecipitation verified interactions for three pairs: Mei-W68_Squ, CSN3_Squ, and Pka-C1_Tej. Furthermore, we screened the majority of *Drosophila* proteins (~12,000) for potential interaction with the Piwi protein, a central player in the piRNA pathway, identifying 164 pairs as potential binding partners. This in silico approach not only efficiently identifies potential interaction partners but also significantly bridges the gap by facilitating the integration of bioinformatics and experimental biology.

## Introduction

Around 10,000–20,000 different types of proteins are encoded in the genome of most organisms, catalyzing the vast majority of physico-chemical reactions in cells (*Galperin et al., 2021*). Many proteins have specialized functions and are often regulated through protein–protein interactions,

where the formation of protein complexes can activate, inhibit, or stabilize their partners. Furthermore, protein–protein interactions can recruit target proteins to specific locations where they will function or regulate the mobility of the protein complex (*Phair and Misteli, 2000*). Within cells, proteins are thought to exist in a crowded environment and frequently interact with other molecules (*Yu et al., 2016*). Thus, characterizing protein–protein interactions is fundamental for understanding protein function and regulation. Large-scale analyses of protein–protein interactions have been carried out, including Tandem Affinity Purification coupled with Mass Spectrometry for the yeast proteome (*Gavin et al., 2002*) and the comprehensive 2-hybrid screening for the Human Reference Interactome (*Luck et al., 2020*). Despite these extensive studies, the overall protein–protein interactions are still not fully understood in many organisms.

The binding between proteins is significantly influenced by their three-dimensional (3D) structures. The characteristics of their interfaces, including hydrogen bonds, salt bridges, and hydrophobicity, determine the interactions (*Keskin et al., 2008*). Therefore, to analyze protein–protein interactions physically and chemically, information on the individual 3D structures of proteins is necessary. The 3D structures of proteins have been determined through experimental methods such as X-ray crystallography, nuclear magnetic resonance (NMR), and cryo-electron microscopy (*Burley et al., 2023*). However, these techniques demand considerable labor and time. The recently developed AlphaFold2 program can predict the 3D structure from its amino acid sequence with high accuracy (*Jumper et al., 2021*). AlphaFold2 requires sequence homology information to predict protein–protein interactions and the complex structure model. The reliability of these predictions is basically dependent on the strength of co-evolutionary signals (*Evans et al., 2021*). This tool has not only been utilized in computational studies but has also become a valuable resource in experimental sciences for predicting protein complexes, as demonstrated with yeast protein complexes (*Humphreys et al., 2021*).

In this study, we attempted a rapid screening of the protein interactions using AlphaFold2 prediction, primarily focusing on components of nuage, a germline-specific, non-membrane organelle that involves a wide variety of proteins containing unique motifs and domains in *Drosophila melanogaster* (*Pek et al., 2012*). Nuage is known to serve as the production and amplification site for small noncoding piRNA, which is bound to PIWI-family proteins. The piRNAs and the PIWI family proteins function to repress mobile genetic elements, or transposons, that disrupt the genomes through their active transpositions (*Ross et al., 2014*). Not only proteins involved in piRNA production, but also translation repressor proteins, including Me31B, Cup, and Trailer hitch (Tral), also localize in nuage (*McCambridge et al., 2020*). Previous studies have shown that the localization of several components in nuage depends on their partners in a hierarchical manner (*Lim and Kai, 2007*). However, the interaction and organization among nuage components remain unclear.

By using AlphaFold2 predictions, we investigated 20 of the nuage-localizing or piRNA-related proteins for pairwise interactions. AlphaFold2 was initially trained to predict the structure of individual proteins (*Jumper et al., 2021*). Its application to complex prediction is an extrapolative use beyond its original intended scope, and its accuracy remains unverified. Even high-confidence predictions may not correspond to actual interactions, necessitating experimental validation to confirm whether predicted protein dimers truly bind. In this study, we confirmed the novel interactions of candidate pairs, including Spindle-E (Spn-E)_Squash (Squ), by co-immunoprecipitation assay using cultured cells. In addition, a Squ mutant, which disrupts the salt bridges predicted at the interface with Spn-E, failed to interact with Spn-E, validating the accuracy of the predicted dimer structure. This screening was expanded for direct interacting pairs between piRNA-related proteins and proteins involved in oogenesis, as well as Piwi and other *Drosophila* proteins. This in silico approach not only streamlines the identification of interaction partners but also bridges the gap between bioinformatics predictions and experimental validation in biological research.

## Results and discussion

### The nuage-localizing proteins and piRNA-related proteins used in the AlphaFold2 screening

Several dozen proteins engaged in piRNA production in germline cells exert their function by recruiting piRNA precursors and interacting with their partner proteins, forming non-membrane structure called a nuage (*Pek et al., 2012*; *Lim and Kai, 2007*). Previous studies reported that many piRNA-related

**Table 1.** The piRNA production-related proteins used in this study.

| Protein | Ortholog | Number of residues | Domain | Direct binding (MIST database) | Localization | Reference |
|---------|----------|--------------------|--------|--------------------------------|--------------|-----------|
| Vas | DDX4 | 661 | DEAD-Box, Hel-C | Aub | Nuage | *Lim and Kai, 2007* |
| Spn-E | Tdrd9 | 1434 | DEAD-Box, Hel-C, HA2, eTud | | Nuage | *Andress et al., 2016* |
| Tej | Tdrd5 | 559 | Lotus, eTud | | Nuage | *Lin et al., 2023* |
| Tapas | Tdrd7 | 1222 | Lotus, eTud | | Nuage | *Patil et al., 2014* |
| Qin | Rnf17 | 1857 | RING, eTud | | Nuage | *Andress et al., 2016* |
| Kots | Tdrd1 | 892 | eTud | | Nuage | *Lim et al., 2022* |
| Krimp | - | 746 | eTud | | Nuage | *Lim and Kai, 2007* |
| Squ | - | 241 | | | Nuage | *Pane et al., 2007* |
| Mael | Mael | 462 | HMG, MAEL | | Nuage | *Lim and Kai, 2007* |
| Aub | PIWIL2 | 866 | N, PAZ, PIWI, MID | Vas, Papi, Me31B | Nuage | *Lim and Kai, 2007* |
| AGO3 | PIWIL4 | 867 | N, PAZ, PIWI, MID | Papi | Nuage | *Webster et al., 2015* |
| Papi | Tdrkh | 576 | eTud, KH | Aub, AGO3 | Mitochondria | *Liu et al., 2011* |
| Vret | Tdrd1 | 691 | eTud | BoYb | Nuage | *Handler et al., 2011* |
| Bel | DDX3 | 801 | DEAD-Box | | Nuage | *Johnstone et al., 2005* |
| Zuc | Pld6 | 253 | PLD-like | Zuc | Mitochondria | *Nguyen et al., 2023* |
| Cup | Eif4enif1 | 1117 | | Me31B | Nuage | *McCambridge et al., 2020* |
| Tral | Lsm14 | 657 | Lsm, FDF | Me31B | Nuage | *McCambridge et al., 2020* |
| Me31B | DDX6 | 459 | DEAD-Box | Aub, Cup, Tral | Nuage | *McCambridge et al., 2020* |
| Shu | Fkbp6 | 455 | PPIase | | Nuage | *Olivieri et al., 2012* |
| BoYb | Tdrd12 | 1059 | DEAD-Box, eTud | Vret | Nuage | *Handler et al., 2011* |

MIST, Molecular Interaction Search Tool.

proteins localized to nuage and some proteins localized in mitochondria (*Table 1*). In addition, protein components of processing bodies and sponge bodies, which are involved in the translation, storage, degradation, and transportation of mRNAs—such as Me31B, Cup, and Tral—also localize to nuage (*McCambridge et al., 2020*; *Table 1*). However, the details of how these proteins interact and organize themselves within the nuage remain unclear.

In this study, we used the AlphaFold2 program to screen for interactions among 20 proteins that are localized in the nuage and/or involved in piRNA production in *Drosophila* (*Table 1*). The monomeric structures of these 20 proteins, ranging in size from 20 kDa to 250 kDa, have already been predicted and are registered in databases (*Varadi et al., 2024*). This set includes both well-structured proteins and those that are largely disordered with numerous loops (*Figure 1—figure supplement 1A*). Of those, eight proteins feature one or more Tudor domains or extended Tudor (eTud) domains. The Tudor domain contains approximately 60 residues and folds into an antiparallel β-sheet with five strands forming a barrel-like fold, while the eTud domains include an additional Oligonucleotide/ oligosaccharide-Binding fold domain (*Ren et al., 2014*). Both Tudor and eTud domains are known to bind predominantly to methylated lysine or arginine residues. In addition, five RNA helicases, such as Vasa (Vas) and the fly homolog of Tdrd9, Spn-E, which are essential for piRNA processing, are also included (*Table 1*). The Vas's C-terminal region is known to bind to the Lotus domain shared by two nuage components, Tejas (Tej) and Tapas. Spn-E is also recently shown to interact with Tej (*Lin et al., 2023*). Among those 20 proteins, the Molecular Interaction Search Tool (MIST), a conventional

database of protein–protein interactions, registers 8 interacting pairs as direct binding, and 28 interactions which are direct or indirect (*Table 1*, *Figure 1—figure supplement 1B and C*; *Hu et al., 2018*).

## Screening for the protein–protein interactions by AlphaFold2

We used AlphaFold2 program to predict the direct protein–protein interaction and 3D structure of the complex. Assuming a 1:1 binding of 20 types of proteins, a total of 400 pairs of dimer predictions were calculated by a supercomputer. The prediction flow of AlphaFold2 consisted of two main parts (*Jumper et al., 2021*). Initially, a multiple sequence alignment was performed for each query protein and stored for the future use. Subsequently, the AlphaFold2 program predicted 3D dimer structures based on the co-evolution inferred from the multiple sequence alignments. For each dimer prediction, five different structure models with varying parameters were generated. Among these, the model with the highest prediction confidence score (ranking confidence) was selected as the final prediction result. The ranking confidence is constituted by two evaluations, the overall structure (pTM) and an evaluation of the dimeric interface (ipTM), emphasizing the interface evaluation as represented by the following formula (*Evans et al., 2021*): ranking confidence = 0.8 × ipTM + 0.2 × pTM.

These three values, ranking confidence, ipTM, and pTM, for each prediction pairs were visualized in the separate heatmaps (*Figure 1A*, *Supplementary file 1*). In general, ranking confidence and ipTM values showed similar trends although a well-structured protein (e.g., Spn-E) tended to have a higher pTM value, which slightly elevated the ranking confidence. Based on this, in this study, we used the ranking confidence as an indicator of the protein–protein interaction. Each heterodimeric pair was calculated twice in the pairwise screening (e.g., proteins A_B and B_A), and the ranking confidences were plotted (*Figure 1B*). The results showed that there was significant variance in the pairs with lower ranking confidences, while pairs with ranking confidences above 0.6 had relatively higher reproducibility. Consequently, we set a threshold of 0.6 and considered protein pairs with ranking confidences above 0.6 as likely complex-forming candidates. This approach identified 13 pairs; seven of these were already known to form complexes, confirming the effectiveness of AlphaFold2 in predicting complex formations (*Table 2*). The highest ranking confidence pair was the Zuc homodimer, possibly because AlphaFold2 had learned from Zuc homodimer's crystal structure registered in the database (*Nishimasu et al., 2012*). The structures of the 20 proteins used in this study have been analyzed to varying extents in previous studies (*Supplementary file 2*). A complex of Vas and the Lotus domain of Osk has been reported (*Jeske et al., 2017*), and based on this complex structure, the interaction between Vas and Tej Lotus domain was predicted with a high score. Although the conformational analyses of the RNA helicase domain and the eTud domain have been reported previously, many of those cover only a subset of the regions and unlikely to affect our predictions in this study.

The predicted 3D structures and the Predicted Aligned Error (PAE) plots for the 12 pairs are shown in *Figure 1C*. Consistent with a previous report using silkworm *Bombyx mori* (*Namba et al., 2022*), both Argonaute 3 (AGO3) and Aub, members of PIWI-family proteins sharing 50–60% amino acid sequence similarity, were predicted to form dimers with Maelstrom (Mael) (*Figure 1Ci and ii*, *Table 2*). AGO3 and Aub appeared well-folded protein except for their N-terminal flexible regions. In contrast, Mael protein was divided into three parts: N-terminal HMG domain, middle MAEL domain, and C-terminal disordered region (*Matsumoto et al., 2015*; *Figure 1Ci and ii*). AlphaFold2 predicted the MAEL domain interacted with AGO3 and Aub.

Me31B, Tral, and Cup are recognized as RNA regulators localized to the nuage and/or sponge body, though they are not directly involved in the piRNA pathway. Previous studies have indicated that these proteins form complexes (*McCambridge et al., 2020*; *Tritschler et al., 2009*; *Tritschler et al., 2008*). Me31B is a well-conserved RNA helicase and showed the tightly folded structure composed of two concatenated RecA helicase domains (*Peter et al., 2019*). On the other hand, Tral and Cup were predicted largely disordered structure with some secondary structures (*Figure 1Ciii and iv*). The predicted dimer structures of Me31B_Tral and Cup_Me31B showed scores of 0.74 and 0.68, respectively. (*Table 2*). Consistent with the previous study (*Tritschler et al., 2008*), AlphaFold2 predicted that the FDF motif of Tral, which contains a Phe-Asp-Phe sequence folded into two a-helixes from residue 405–537, was associated with Me31B (*Figure 1Ciii*). In addition, an α-helix and loop regions of Cup were predicted to make a contact with Me31B (*Figure 1Civ*). BoYb and Vret, both are eTud domain containing proteins (*Chen et al., 2011*) and their direct interaction has been suggested by the high retrieval rate for BoYb in the immunoprecipitant of Vret from the ovary (*Handler et al., 2011*).

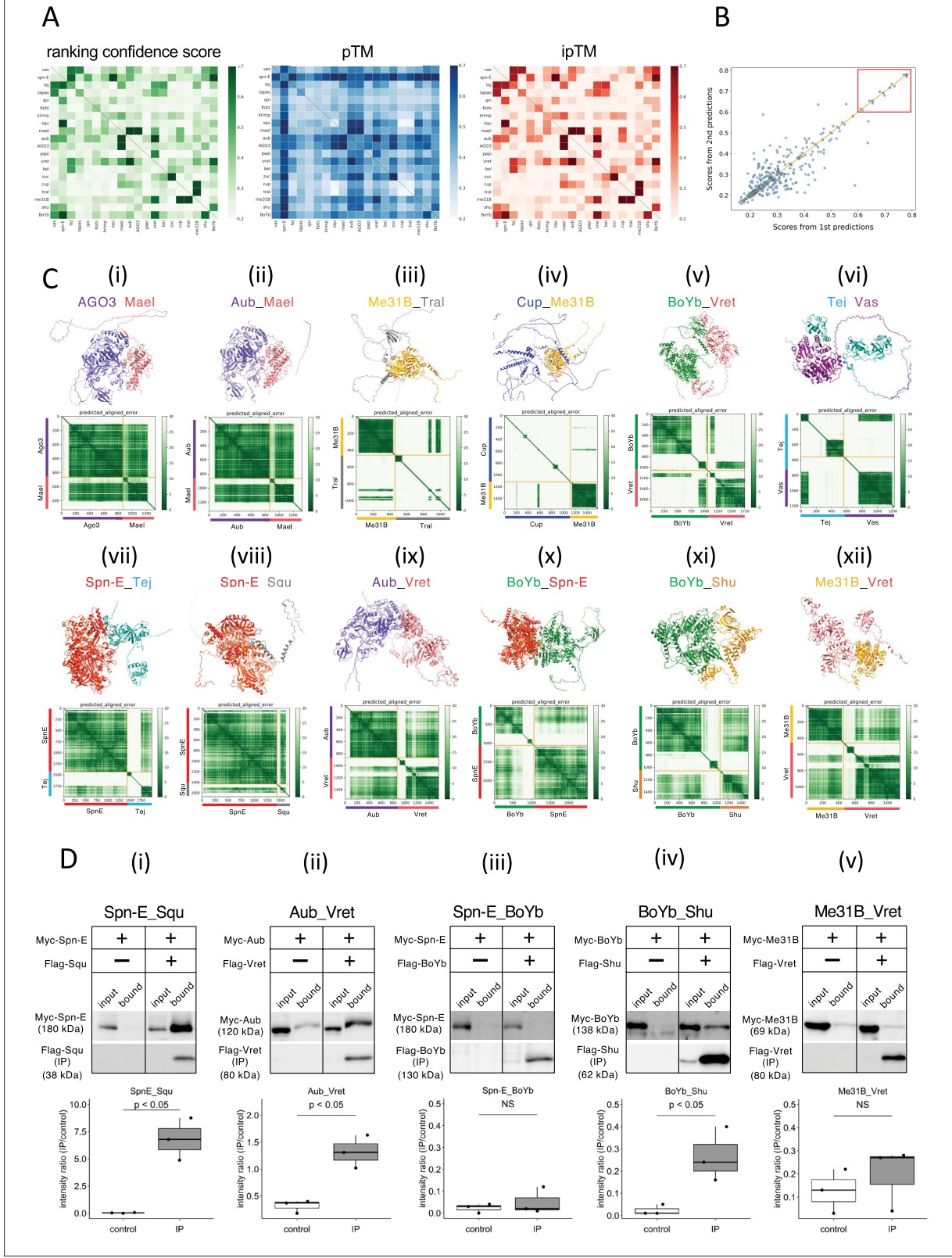

**Figure 1.** The 1:1 dimer structure prediction by AlphaFold2 for piRNA-related proteins. (**A**) Heatmaps of the prediction confidence scores (ranking confidence, green), pTM values (blue), and ipTM values (red) provided by AlphaFold2. The 20 types of proteins are aligned from top to bottom and left to right in the same order. Boxes on diagonal line represent homodimers. (**B**) Scatter plot of the ranking confidences. The scores from first and second predictions for each heterodimer pair are plotted on X and Y axis, respectively. (**Ci~xii**) The predicted 3D structures (top panels) and the Predicted

*Figure 1 continued*

Aligned Error (PAE) plots (bottom panels) for each candidate heterodimers scoring above 0.6. The PAE plot displays the positional errors between all amino acid residue pairs, formatted in a matrix layout. (**D**) Co-immunoprecipitation assays using tagged proteins to verify interactions between specific pairs: Spn-E_Squ (**i**), Aub_Vret (**ii**), Spn-E_BoYb (**iii**), BoYb_Shu (**iv**), and Me31B_Vret (**v**). Single transfected cells expressing only Myc-tagged but not Flag-tagged proteins are used as negative controls for each set. Box and whisker plots show the intensity ratio between immunoprecipitated and input bands (n = 3 biological replicates). p-values were calculated using Student's *t*-test.

The online version of this article includes the following source data and figure supplement(s) for figure 1:

**Source data 1.** PDB files used in *Figure 1C*.

**Source data 2.** Western blots indicating the relevant bands for *Figure 1Di*.

**Source data 3.** Original western blots for *Figure 1Di*.

**Source data 4.** Western blots indicating the relevant bands for *Figure 1Dii*.

**Source data 5.** Original western blots for *Figure 1Dii*.

**Source data 6.** Western blots indicating the relevant bands for *Figure 1Diii*.

**Source data 7.** Original western blots for *Figure 1Diii*.

**Source data 8.** Western blots indicating the relevant bands for *Figure 1Div*.

**Source data 9.** Original western blots for *Figure 1Div*.

**Source data 10.** Western blots indicating the relevant bands for *Figure 1Dv*.

**Source data 11.** Original western blots for *Figure 1Dv*.

**Figure supplement 1.** The nuage proteins analyzed in this study.

The predicted structure revealed that both BoYb and Vret proteins consist of two domains, one at the N-terminal and the other at the C-terminal, connected by a flexible region. (*Figure 1Cv*). Interactions were predicted between their N-terminal domains and between C-terminal domains, respectively. It has been reported that Tej, known as Tdrd5 in mammal, binds directly to Vas through its N-terminal Lotus domain (*Jeske et al., 2017*; *Figure 1Cvi*) and to Spn-E through its loop region continuing the eTud domain (*Lin et al., 2023*; *Figure 1Cvii*). The predicted structures of Tej_Vas and Spn-E_Tej were consistent to their binding properties reported previously.

The remaining five pairs, previously unreported as directly interacting, were considered novel binding pairs (*Table 2*, *Figure 1Cviii–xii*). These interactions were experimentally examined using *Drosophila* S2 culture cells derived from embryonic somatic cells that lack germline-specific proteins. Previously, Squ was co-immunoprecipitated with Spn-E along with other nuage components from

**Table 2.** The screening for the interacting proteins (prediction confidence score, ranking confidence >0.6).

| Protein A_B first prediction | ranking confidence | Protein B_A second prediction | Ranking confidence | Reference | Validation by co-IP |
|---|---|---|---|---|---|
| Zuc_Zuc | 0.85 | N/A | N/A | *Nishimasu et al., 2012* | N/A |
| AGO3_Mael | 0.78 | Mael_AGO3 | 0.78 | *Namba et al., 2022* | N/A |
| Aub_Mael | 0.78 | Mael_Aub | 0.78 | *Namba et al., 2022* | N/A |
| Spn-E_Squ | 0.77 | Squ_Spn-E | 0.78 | This study | ++ |
| Me31B_Tral | 0.74 | Tral_Me31B | 0.72 | *McCambridge et al., 2020* | N/A |
| Aub_Vret | 0.72 | Vret_Aub | 0.72 | This study | + |
| BoYb_Spn-E | 0.69 | Spn-E_BoYb | 0.69 | This study | - |
| Cup_Me31B | 0.68 | Me31B_Cup | 0.70 | *McCambridge et al., 2020* | N/A |
| Spn-E_Tej | 0.65 | Tej_Spn-E | 0.65 | *Lin et al., 2023* | N/A |
| BoYb_Vret | 0.64 | Vret_BoYb | 0.65 | *Handler et al., 2011* | N/A |
| BoYb_Shu | 0.64 | Shu_BoYb | 0.56 | This study | + |
| Me31B_Vret | 0.64 | Vret_Me31B | 0.45 | This study | - |
| Tej_Vas | 0.61 | Vas_Tej | 0.62 | *Patil and Kai, 2010* | N/A |

ovarian lysate (*Andress et al., 2016*), but whether this interaction was direct had not been examined. Co-immunoprecipitation assay in S2 cells, Myc-Spn-E was strongly detected in the precipitant of Flag-Squ by western blotting, possibly supporting the direct interaction between Spn-E and Squ in the S2 cells devoid of germline proteins (*Figure 1Di*). Similarly, AlphaFold2 predicted a direct interaction between Aub and Vret, which was corroborated by co-immunoprecipitation assays (*Figure 1Dii*). The binding capabilities of another pair, BoYb-Shutdown (Shu), were also confirmed in S2 cells (*Figure 1Div*). Three out of five candidate pairs confirmed interactions, validating the effectiveness of AlphaFold2 in identifying the binding partners. However, BoYb-Spn-E and Me31B-Vret did not show interaction in these assays (*Figure 1Diii and v*), possibly suggesting weak interactions that co-immunoprecipitation may have failed to detect. While co-immunoprecipitation is a widely used method, it may not always detect weak or transient interactions. Other validation methods, such as FRET or co-localization assay in culture cells, could offer further insights to support the results. It is also important to note that AlphaFold2's predictions are not definitive and may lead to false positives, particularly when analyzing a large number of interactions.

## Evaluation of Spn-E and Squ interaction in culture cells and ovaries

Among the binding candidates, we focused on the predicted dimer structure of Spn-E and Squ pair. Spn-E is an evolutionarily conserved RNA helicase that is expressed in germline cells. It plays a crucial role in the piRNA production and transposon suppression in germline cells (*Andress et al., 2016*; *Czech et al., 2013*). Similarly, Squ is also expressed in ovary and testis and involved in the piRNA production, although its molecular role is less defined (*Czech et al., 2013*; *Pane et al., 2007*). While *squ* is conserved across *Drosophila* species (*Figure 2—figure supplement 1A, B*), vertebrate orthologs remain unidentified. Spn-E contains four domains: DEAD/DEAH helicase, Hel-C, HA2, and eTud domains (*Figure 2A*). Its predicted 3D structure was well folded and contained few flexible regions (*Figure 1Cviii*). In contrast, Squ was predicted to be largely disordered, consisting of three α-helices and two β-strands (*Figure 2A*). The middle parts of Squ were in close contact with Spn-E, showing lower PAE values, suggestive of their interaction (*Figures 1Cviii and 2A*). AlphaFold2 predicts the five structure models for each query using different initial model parameters (models 1–5) and ranking confidence is given to each model. As for Spn-E_Squ pair, the ranking confidence scores were ranging from 0.74 to 0.77. The 3D structures of Spn-E were very similar across all five models, superimposing almost perfectly (*Figure 2B*). The middle region of Squ was consistently positioned relative to Spn-E, although the N- and C-terminal regions of Squ remained flexible (*Figure 2B*).

The closer examination of the Spn-E_Squ dimer interface revealed a short α-helix of Squ (106th–116th residues) fitted into a groove on the Spn-E surface, while the anti-parallel β-sheet (140th–153rd) was also predicted to interact with Spn-E (*Figure 2A and C*). Physico-chemical structural analysis using PDBePISA server (EMBL-EBI) identified salt bridges between Spn-E and Squ (*Supplementary file 3*; *Supplementary file 4*; *Krissinel and Henrick, 2007*). To validate these predicted interactions, we generated Squ mutants substituting each residue involved in the four salt bridges (E107, E109, R115, and K163) with alanine (*Figure 2D*, *Figure 2—figure supplement 1B*) and assessed their interactions by co-immunoprecipitation in S2 cells expressing tagged proteins, Myc-Spn-E and Flag-Squ. The assay revealed that while the E107A single mutation did not affect the interaction, other single mutations mildly reduced the binding affinity of Squ to Spn-E (*Figure 2—figure supplement 2A*), Furthermore, the localization of GFP-tagged Squ and mKate2 (mK2)-tagged Spn-E was examined in S2 cells. When only Squ was expressed, it was dispersed in cytosol (*Figure 2—figure supplement 2B*). On the other hand, when only Spn-E was expressed, it localized in the nucleus as reported previously (*Lin et al., 2023*). In the co-expression of Squ wildtype or single mutants, Spn-E was moved to the cytoplasm and form granules together with Squ, suggesting the interaction between them. Although the single mutants still could bind to Spn-E, Squ quadruple mutant (Squ[4A]) completely lost the binding (*Figure 2E*) and did not show the co-localization with Spn-E in S2 cells (*Figure 2—figure supplement 2B*). These results suggest that the salt bridges are important for the interaction between Spn-E and Squ and support the accuracy of their dimer structure predicted by AlphaFold2.

While the RNA binding site of Spn-E has not been extensively studied, it is presumed to be near the helicase domain, similar to the Vas helicase-RNA complex (*Sengoku et al., 2006*). In addition, *Lin et al., 2023* demonstrated that Hel-C domain of Spn-E interacted with the Tej's eSRS region, which recruits Spn-E to nuage, a site distinct from the predicted Squ binding sites (*Figure 2A*). Interestingly, a

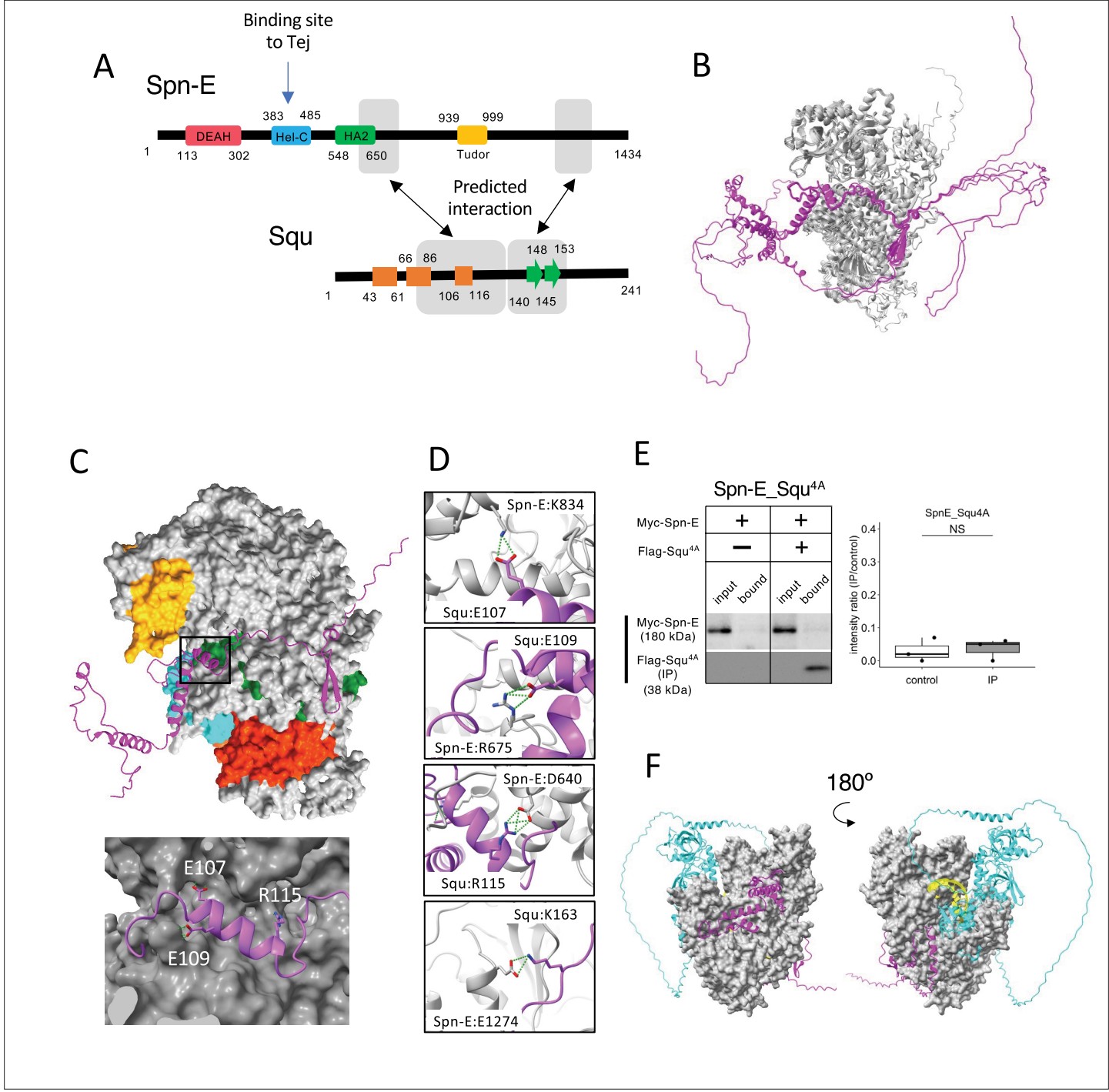

**Figure 2.** Interaction between Spn-E and Squ. (**A**) Schematic of Spn-E domain structures defined in SMART (**Letunic et al., 2021**). Boxes (α-helix: orange) and arrow (β-sheet: green) for Squ structure. The predicted interacting regions between Spn-E and Squ are indicated in gray boxes. Tej interaction site of Spn-E is also shown (**Lin et al., 2023**). (**B**) The predicted five models of heterodimer of Spn-E (in gray) and Squ (in magenta). Spn-E molecules in all five models are superimposed. (**C**) 3D structure of the Spn-E_Squ dimer colored by Spn-E domains as indicated in (**A**), with Squ in magenta. The enlarged image of the interface indicated by box is also shown. (**D**) The predicted salt bridges at the interface, with Spn-E in gray and Squ in magenta. The residues forming salt bridges are depicted in stick model. (**E**) Co-immunoprecipitation assay using S2 cell lysate to examine the interaction between Myc-Spn-E and Flag-Squ mutant (4A) whose salt bridge-forming residues are mutated to Ala. S2 cells expressing Myc-Spn-E alone is used as a control. The ratios of the band intensity (IP/input) are shown in a box and whisker plot (n = 3 biological replicates). p-values were calculated using Student's t-test. (**F**) The heterotetramer model of Spn-E_Squ_Tej_RNA predicted by AlphaFold3. Spn-E is shown as a space filled model in gray, Squ in magenta, Tej in cyan, and RNA in yellow. The model on the left is rotated 180° in the Y axis to produce the image on the right.

*Figure 2 continued on next page*

*Figure 2 continued*

The online version of this article includes the following source data and figure supplement(s) for figure 2:

**Source data 1.** PDB files used in *Figure 2B*.

**Source data 2.** Western blots indicating the relevant bands for *Figure 2E*.

**Source data 3.** Original western blots for *Figure 2E*.

**Source data 4.** CIF file used in *Figure 2F*.

**Figure supplement 1.** Comparative analysis of Squ and Spn-E orthologs in *Drosophila.*

**Figure supplement 2.** Interaction and localization analysis of Spn-E and Squ in S2 cells.

**Figure supplement 2—source data 1.** Western blots indicating the relevant bands for *Figure 2—figure supplement 2A*.

**Figure supplement 2—source data 2.** Original western blots for *Figure 2—figure supplement 2A*.

**Figure supplement 2—source data 3.** Confocal microscopy images in *Figure 2—figure supplement 2B*.

**Figure supplement 3.** Trimer structures predicted by AlphaFold3.

**Figure supplement 3—source data 1.** CIF files used in *Figure 2—figure supplement 3A–D*.

tetramer complex of Spn-E_Squ_Tej_RNA predicted by the recently available AlphaFold3 (*Abramson et al., 2024*) placed the single-strand RNA (ssRNA) near Spn-E's helicase domain (*Figure 2F*), aligning with the ssRNA binding position found in Vas (*Figure 2—figure supplement 2C*). The predicted tetramer model suggests that Squ binding to Spn-E does not inhibit but may potentially regulate Spn-E's interaction with Tej or RNA by stabilizing the domain orientation of Spn-E (*Figure 2F*).

In addition to the Spn-E_Squ_Tej complex, 1:1 dimer prediction described above further suggested potential trimers (*Figure 1*, *Figure 2—figure supplement 3*). For example, Tej protein is predicted to bind both Vas and Spn-E, and AlfaFold3 indeed further predicted a Vas_Tej_Spn-E trimer, where Tej's Lotus and eTud domains interact with Vas and Spn-E, respectively. However, Lin et al. reported that Tej binds exclusively either with Vas or Spn-E, but not simultaneously, in *Drosophila* ovary (*Lin et al., 2023*), suggesting that the predicted trimers may be weak or transient. Similarly, the BoYb_Vret_Shu and the Me31B_Cup_Tral trimers remain hypothetical and require experimental verification (*Figure 2—figure supplement 3*).

We investigated whether Spn-E also interacts with Squ within the *Drosophila* ovary. The antibody against Squ detected a specific band at the expected size by western blotting in the heterozygous control ovarian lysate, which was absent in the transheterozygote mutant, $squ^{PP32/HE47}$ (*Figure 3A*; *Pane et al., 2007*). Consistent with the previous report conducted with the transgenic line expressing HA-Squ (*Pane et al., 2007*), immunostaining of ovaries revealed the Squ's localization in nuage, which overlaps with endogenously-tagged Spn-E with mK2 (*Figure 3B*). Spn-E was co-immunoprecipitated together with Squ from ovarian lysate, indicating the interaction between Squ and Spn-E (*Figure 3C*). While the previous mass spectrometry analysis detected PIWI family proteins, Piwi, Aub, and AGO3, in Spn-E immunoprecipitates (*Andress et al., 2016*), these three proteins were not present in the immunoprecipitant of Squ (*Figure 3C*), further supporting the direct interaction between Squ and Spn-E.

In this study, three novel protein–protein interactions were predicted and experimentally confirmed. AlphaFold2 also predicted the 3D structure of these complexes, providing insight into the important regions involved in complex formation. These predictions will provide fundamental information to elucidate nuage assembly. Nuage is thought to form by liquid-phase separation; however, direct protein–protein interactions likely occur within protein-dense nuage, facilitating RNA processing. Although the precise roles of individual interactions require further study, characterization of protein–protein interactions within nuage will help clarify the mechanism of piRNA production.

## Screening oogenesis-related proteins for interaction with nuage proteins

Given the role of nuage for piRNA biogenesis and germline development, interactions between nuage-localized proteins and those involved in oogenesis were expected. We employed AlphaFold2 to predict these interactions using Vas, Squ, and Tej, the representative nuage components yet remain elusive, as baits. Of 430 proteins in oogenesis pathway (*Aleksander et al., 2023*), dimeric binding of 1290 pairs was predicted (*Supplementary file 5*), with 18 pairs showing dimer structures scoring above 0.6 (*Table 3*). Among those, co-immunoprecipitation in S2 cells confirmed interactions of three

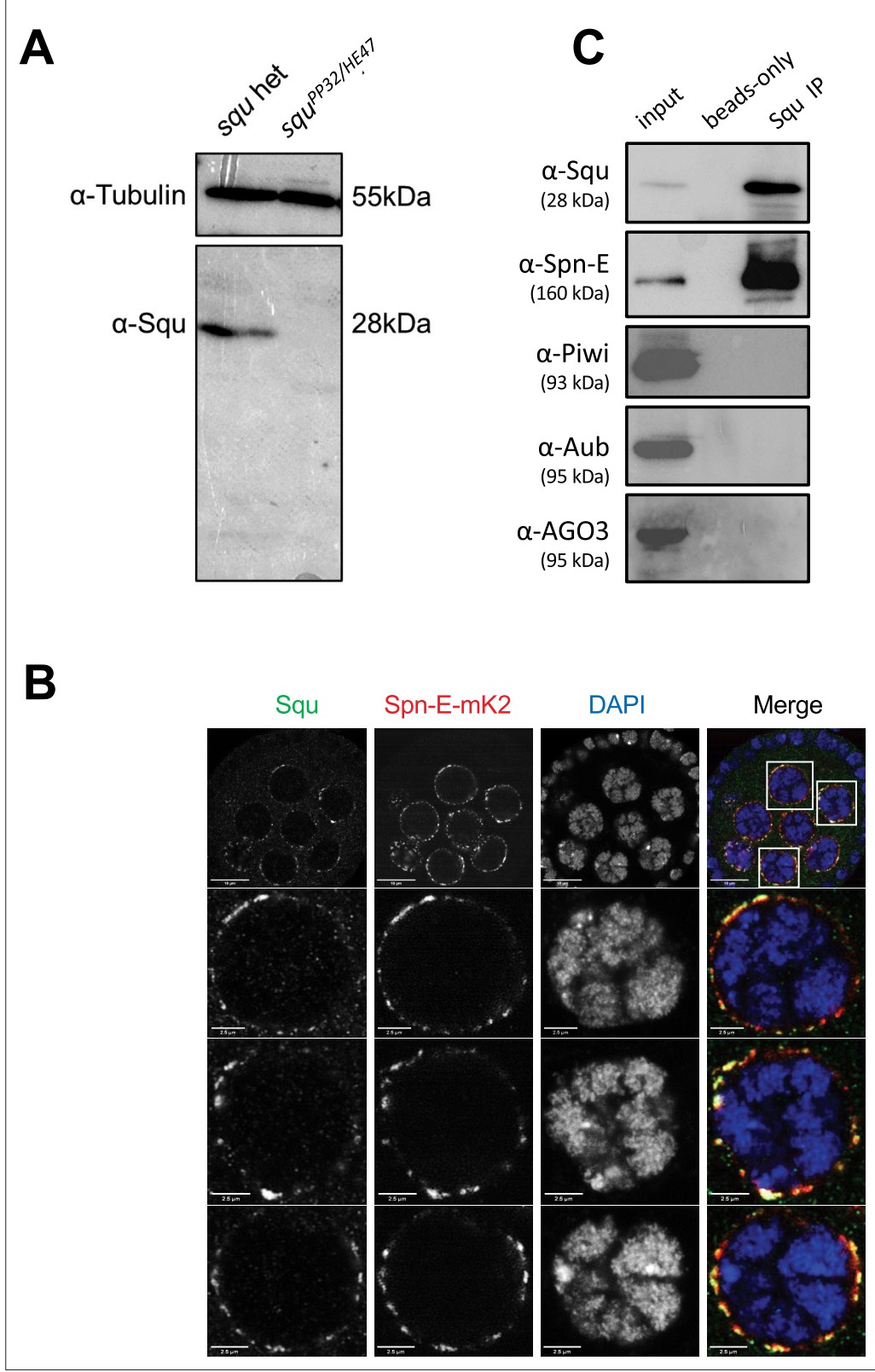

**Figure 3.** Spn-E and Squ interact in *Drosophila* ovary. (**A**) Western blotting analysis using anti-Squ antibody reveals a specific band at the expected size (approximately 28 kDa) for endogenous Squ in *Drosophila* ovarian lysates of the heterozygous control. This band is absent in the transheterozygote, *squ^{PP32/HE47}*. (**B**) Immunostaining of *Drosophila* egg chambers with anti-Squ antibody and anti-mKate2 (mK2) antibody demonstrates colocalization

*Figure 3 continued on next page*

*Figure 3 continued*

of Squ and Spn-E-mK2 in nuage, a perinuclear granule in germline cells. The enlarged images of nuclei are shown in the panels below. Scale bars: 10 μm (top row), 2.5 μm (enlarged images). (**C**) Immunoprecipitation of the endogenous Squ from ovarian lysate revealed the interaction with Spn-E protein. Proteins were detected by western blotting analysis using the specific antibody for each protein. The negative control was performed without anti-Squ antibody (beads only).

The online version of this article includes the following source data for figure 3:

**Source data 1.** Western blot indicating the relevant bands for *Figure 3A*.

**Source data 2.** Original western blot for *Figure 3A*.

**Source data 3.** Confocal microscopy images in *Figure 3B*.

**Source data 4.** Western blots indicating the relevant bands for *Figure 3C*.

**Source data 5.** Original western blots for *Figure 3C*.

---

pairs, Mei-W68_Squ, CSN3_Squ, and Pka-C1_Tej (*Figure 4A and B*, *Table 3*). The Mei-W68_Squ dimer, scoring 0.63, the binding site of Squ to Mei-W68 was predicted at α-helixes in its middle region, which overlaps with the interacting site to Spn-E (*Table 3*, *Figure 4Ai*, compare with *Figure 1Cviii*). Mei-W68 is a topoisomerase, known as Spo11 in many organisms, which is required for the formation of double-strand breaks during meiosis (*McKim and Hayashi-Hagihara, 1998*). Interestingly, Squ also plays a role in DNA damage response pathway and showed the genetic interaction with *chk2*, a meiotic checkpoint gene (*Pane et al., 2007*). These results suggest that the binding of Squ to Mei-W68 may regulate the enzymatic activity of Mei-W68 in order to suppress the excessive formation of double-strand breaks. Another confirmed pair was CSN3_Squ pair scoring 0.62 (*Figure 4Aii and Bii*). CSN3, a component of COP9 signalosome which removes Nedd8 modifications from target

---

**Table 3.** The binding candidates predicted by AlphaFold2.

| Protein_A | Protein_B | AlphaFold2 ranking confidence | Validation by co-IP | Function of Protein_A |
|---|---|---|---|---|
| Vps25 | Squ | 0.71 | No | A member of the ESCRT-II complex |
| Nup44A | Squ | 0.65 | No | A nuclear pore protein |
| Nclb | Squ | 0.64 | No | Chromatin-associated factor |
| Mei-W68 | Squ | 0.63 | Bound | Formation of double-strand breaks |
| DNaseII | Squ | 0.63 | N/E | Deoxyribonuclease II |
| Spn-D | Squ | 0.62 | No | Homologous recombinational DNA repair |
| CSN3 | Squ | 0.62 | Bound | Subunit of the COP9 signalosome |
| Jagn | Tej | 0.72 | No | Located in the endoplasmic reticulum |
| Pka-C1 | Tej | 0.64 | Bound | Serine/threonine kinase |
| Rab7 | Tej | 0.62 | No | Vesicle trafficking regulation |
| Baf | Vas | 0.85 | No | Chromatin organization |
| Mats | Vas | 0.79 | No | Coactivator of Warts (Wts) kinase |
| Abo | Vas | 0.68 | No | Negative regulator of histone transcription genes |
| CathD | Vas | 0.67 | N/E | Apoptosis and the defense response |
| Rab11 | Vas | 0.67 | No | Endomembrane trafficking |
| Vls | Vas | 0.63 | No | Substrate recognition platform for cusl |
| Hsc70-4 | Vas | 0.62 | No | Protein folding |
| RhoL | Vas | 0.61 | N/E | Maturation of hemocytes |

The expression plasmids were not constructed due to the technical reasons.

N/E, not examined.

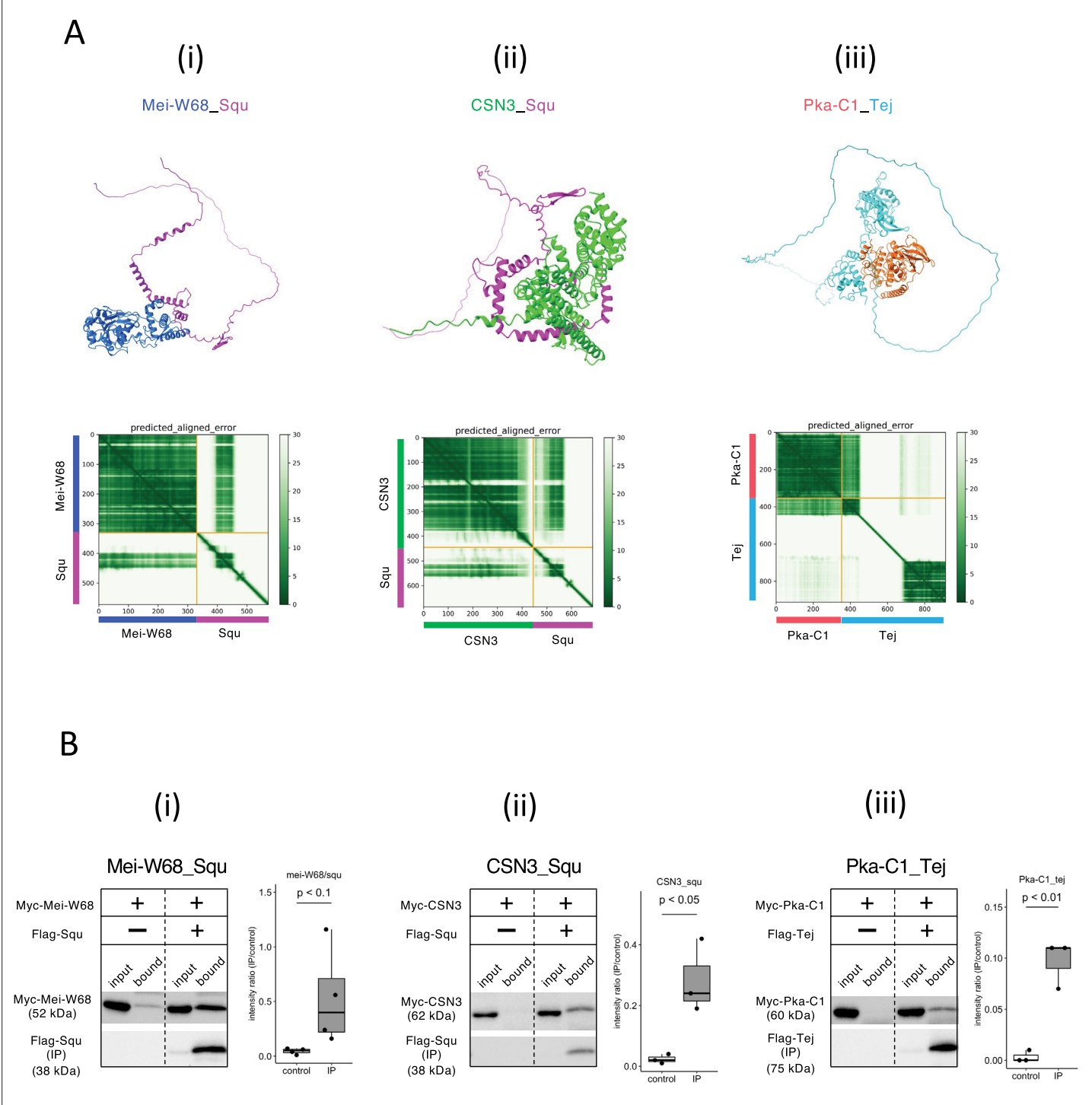

**Figure 4.** Squ- and Tej-interacting proteins predicted by AlphaFold2. (**Ai–iii**) The predicted dimer structures (top) and Predicted Aligned Error (PAE) plots (bottom) of Mei-W68 in blue and Squ in magenta (**i**), CSN3 in green and Squ in magenta (**ii**), Pka-C1 in orange and Tej in cyan (**iii**). The PAE plot displays the positional errors between all amino acid residue pairs, formatted in a matrix layout. (**Bi–iii**) Co-immunoprecipitation assays using tagged proteins to verify interactions between specific pairs: Mei-W68_Squ (**i**), CSN3_Squ (**ii**), and Pka-C1_Tej (**iii**). Single transfected cells expressing only Myc-tagged but not Flag-tagged proteins are used as negative controls for each set. Box and whisker plots show the intensity ratio between immunoprecipitated and input bands (n = 3 biological replicates). p-values were calculated using Student's *t*-test.

The online version of this article includes the following source data and figure supplement(s) for figure 4:

**Source data 1.** PDB files used in *Figure 4A*.

*Figure 4 continued on next page*

*Figure 4 continued*

**Source data 2.** Western blots indicating the relevant bands for *Figure 4Bi*.

**Source data 3.** Original western blots for *Figure 4Bi*.

**Source data 4.** Western blots indicating the relevant bands for *Figure 4Bii*.

**Source data 5.** Original western blots for *Figure 4Bii*.

**Source data 6.** Western blots indicating the relevant bands for *Figure 4Biii*.

**Source data 7.** Original western blots for *Figure 4Biii*.

**Figure supplement 1.** Validation of predicted protein interactions via co-immunoprecipitation from S2 cell lysate.

**Figure supplement 1—source data 1.** Western blots indicating the relevant bands for *Figure 4—figure supplement 1A*.

**Figure supplement 1—source data 2.** Original western blots for *Figure 4—figure supplement 1A*.

**Figure supplement 1—source data 3.** Western blots indicating the relevant bands for *Figure 4—figure supplement 1B*.

**Figure supplement 1—source data 4.** Original western blots for *Figure 4—figure supplement 1B*.

**Figure supplement 1—source data 5.** Western blots indicating the relevant bands for *Figure 4—figure supplement 1C* (Abo, Baf, Hsc70-4).

**Figure supplement 1—source data 6.** Western blots indicating the relevant bands for *Figure 4—figure supplement 1C* (Mats, Rab11, Vls).

**Figure supplement 1—source data 7.** Original western blots for *Figure 4—figure supplement 1C*.

proteins, is required for the self-renewal of the germline stem cells (*Pan et al., 2014*). Pka-C1, a cAMP-dependent protein kinase involved in axis specification, rhythmic behavior and synaptic transmission (*Öztürk-Çolak et al., 2024*) and predicted to bind with the N-terminal Lotus domain of Tej (Score 0.64, *Figure 4Aiii and Biii*), which is also known as binding site to Vas (*Jeske et al., 2017*). This suggests a potential competitive interaction between Pka-C1 and Vas for Tej. Although the success rate of confirmed interactions was low (3 out of 18) (*Table 3*, *Figure 4—figure supplement 1*), the results indicate that these protein pairs could interact within cells if co-expressed in vivo. The ranking confidence score reflects the reliability of AlphaFold2's predicted structure but does not always ensure accuracy. Therefore, we assessed complex affinity based on the predicted three-dimensional structures (*Xue et al., 2016*; *Supplementary file 6*). Most dimers with high-ranking confidence scores exhibited low Kd values indicative of high affinity, while some showed high Kd values indicating weak interactions (*Supplementary file 6*). For example, the Baf_Vas complex had a high AlphaFold2 ranking confidence score (0.85) but a relatively high Kd value (1.1E-4 M), indicating low affinity. Consistently, Baf_Vas binding was not detected in co-IP experiments (*Figure 4—figure supplement 1C*). Although accurate Kd prediction may be limited due to insufficient structural optimization, it could serve as a valuable secondary screening tool following AlphaFold2 predictions.

## Screening all *Drosophila* proteins for Piwi-interacting proteins

Given the crucial role of Piwi in piRNA biogenesis, heterochromatin formation, and germline stem cell (GSC) maintenance, we employed AlfaFold2 to screen all proteins in *D. melanogaster* for potential Piwi interactions. Piwi, the founder member of the PIWI family proteins, is not only essential for binding piRNAs and regulating complementary mRNAs but also plays a critical role in GSC self-renewal (*Klenov et al., 2011*). Studies have shown that Piwi, lacking the N-terminal moiety containing the nuclear localization signal (NLS), still retains GSC self-renewal capabilities. Its function in GSC self-renewal is realized independently in the cytoplasm of GSC niche cells, separate from its role in transposon repression. The crystal structures of *Drosophila* Piwi and silkworm Siwi have been solved and revealed the organization of four domains (N, PAZ, MID, and PIWI) (*Matsumoto et al., 2016*; *Yamaguchi et al., 2020*). Recently, the ternary structure of piRNA, target RNA, and MILI, a mouse ortholog of Piwi, has been reported and the bound piRNA threaded through the channel between N-PAZ and MID–PIWI lobes (*Figure 5—figure supplement 1A*; *Li et al., 2024*).

To identify novel Piwi-binding proteins, we conducted a 1:1 interaction screening involving approximately 12,000 *Drosophila* proteins, excluding any proteins over 2000 amino acid residues due to the computational limits. The ranking confidences by AlphaFold2 were primarily low, with over 98% being below 0.6, suggesting a low likelihood of interaction between Piwi and the vast majority of the proteins (*Figure 5A*). Approximately 1.5% of the pairs, totaling 164 pairs, scored above 0.6, was expected to contain the novel binding partners (*Supplementary file 7*). Top 24 candidates with greater than 0.75 ranking confidence are listed in *Table 4*. This list contained many metabolic enzymes

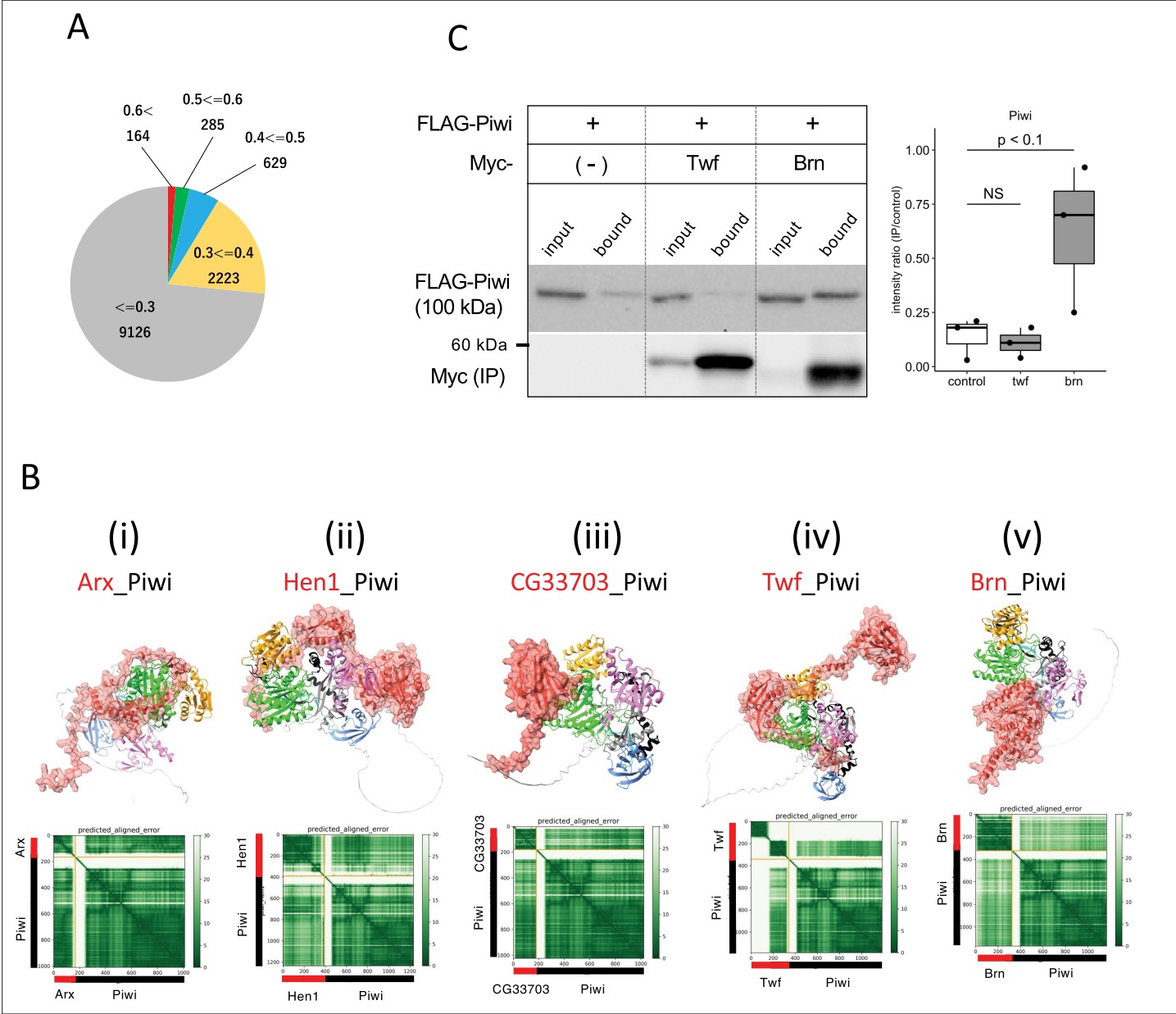

**Figure 5.** Screening for Piwi-interacting proteins in *Drosophila* proteome. (**A**) Pie chart displaying the distribution of ranking confidences from the AlphaFold2 screening for Piwi-interacting proteins among those encoded by *Drosophila* genome. (**Bi–v**) The predicted dimer structure (top) and PAE plots (bottom) for the Piwi and the binding candidates in red: Arx (**i**), Hen1 (**ii**), CG33703 (**iii**), Twf (**iv**), and Brn (**v**). Piwi is shown in the same colors as *Figure 5—figure supplement 1A*. (**C**) Co-immunoprecipitation assays using tagged proteins to verify interactions between Piwi and the binding candidates, Twf and Brn. Single transfected cells expressing only Flag-Piwi is used as negative control. Box and whisker plots show the intensity ratio between immunoprecipitated and input bands (n = 3 biological replicates). p-values were calculated using Student's *t*-test.

The online version of this article includes the following source data and figure supplement(s) for figure 5:

**Source data 1.** PDB files used in *Figure 5B*.

**Source data 2.** Western blots indicating the relevant bands for *Figure 5C*.

**Source data 3.** Original western blots for *Figure 5C*.

**Figure supplement 1.** Structural analyses of Piwi complexes and interactions.

**Table 4.** Piwi-interacting proteins predicted by AlphaFold2 (score ≥ 0.75).

| Protein | Length (residue) | Ranking confidence | Human ortholog | Gene summary (FlyBase) |
|---|---|---|---|---|
| CG34283 | 153 | 0.85 | GTSF1 | - |
| CG32625 | 144 | 0.84 | GTSF1 | - |
| Arx | 167 | 0.83 | GTSF1 | It plays an essential role in piRNA-guided transcriptional silencing, interacting probably directly with the product of piwi |
| CG33703 | 181 | 0.82 | - | No phenotypic data is available |
| GstE12 | 223 | 0.82 | GSTT2B | Glutathione S transferase E12 (GstE12) encodes an enzyme involved in glutathione metabolism |
| CAH4 | 279 | 0.81 | CA6 | Predicted to enable carbonate dehydratase activity. Predicted to be active in cytoplasm |
| CG13192 | 323 | 0.81 | GNB1L | Predicted to be involved in social behavior |
| Mael | 462 | 0.79 | MAEL | Involved both in the piRNA and miRNA metabolic processes |
| Adk3 | 366 | 0.78 | ADK | Predicted to enable adenosine kinase activity |
| Alg11 | 475 | 0.78 | ALG11 | Predicted to enable GDP-Man:Man3GlcNAc2-PP-Dol alpha-1,2-mannosyltransferase activity |
| CG41378 | 228 | 0.78 | IFI30 | Predicted to enable oxidoreductase activity |
| CG14036 | 93 | 0.77 | GTSF1 | Involved in copper ion homeostasis |
| CG7966 | 486 | 0.77 | SELENBP1 | Predicted to enable methanethiol oxidase activity |
| Hen1 | 391 | 0.77 | HENMT1 | Hen1 encodes a methyltransferase that methylates the terminal 2' hydroxyl group of small interfering RNAs and Piwi-interacting RNAs |
| Rpp14b | 112 | 0.77 | RPP14 | Predicted to enable ribonuclease P RNA binding activity |
| CG33783 | 164 | 0.76 | - | No phenotypic data is available |
| AANATL4 | 224 | 0.75 | - | Predicted to enable aralkylamine N-acetyltransferase activity |
| CG14787 | 260 | 0.75 | CDYL2 | Is expressed in adult heart; embryonic Malpighian tubule; and embryonic main segment of Malpighian tubule |
| CG33160 | 258 | 0.75 | PRSS1 | Predicted to enable serine-type endopeptidase activity |
| CG3397 | 342 | 0.75 | AKR7A2 | Predicted to enable D-arabinose 1-dehydrogenase [NAD(P)+] activity |
| CG4390 | 330 | 0.75 | ESD | Enables serine hydrolase activity |
| CG7142 | 334 | 0.75 | KLK1 | Predicted to enable serine-type endopeptidase activity |
| JanA | 135 | 0.75 | PHPT1 | JanA and janB regulate somatic sex differentiation |
| Yip7 | 270 | 0.75 | CTRB1 | Enables serine hydrolase activity |

and three piRNA-related proteins, Asterix (Arx), Mael, and Hen1. The interactions between Mael and Piwi-family proteins have been already reported (*Namba et al., 2022*). Arx, known as Gtsf1 in mammals and integral to Piwi–piRISC-mediated transcriptional silencing in nucleus (*Ohtani et al., 2013*), had high ranking confidences (0.83, *Table 4*). Despite its known three-dimensional structure determined by NMR spectroscopy (*Ipsaro et al., 2021*), the Arx_Piwi complex structure remained elusive. AlphaFoldF2 predicted that while Arx lacked a compact domain, the majority of Arx protein associated around the PIWI domain, except for the flexible C-terminal region (130th–167th residues) (*Figure 5Bi*). Three Arx paralogs in *Drosophila* (CG34283, CG32625, and CG14036) were also predicted to bind to Piwi with high-ranking confidences, suggesting their interactions within the cells (*Figure 5—figure supplement 1B*). Although CG34283 is not expressed, CG32625 and CG14036 are moderately and highly expressed in ovary, respectively (*Öztürk-Çolak et al., 2024*). However, unlike *arx*, knockdown of each paralogous gene did not result in de-repression of a transposon, *mdg1* (*Ohtani et al., 2013*), suggesting that they may be pseudogenes or possess redundant roles.

Hen1 is a methyltransferase known to mediate methylation of the terminal 2' hydroxyl group of small interfering RNAs and piRNAs, thereby enhancing the stability of the small RNAs. Consistent with the previous report showing Hen1 binding to Piwi (*Ohtani et al., 2013*), the dimer structure of Hen1_Piwi was predicted with high-ranking confidence, 0.77. This prediction further suggests that Hen1 is recruited to Piwi, thereby positioning it closer to the piRNA substrate (*Figure 5Bii*). Another potential interacting protein for Piwi was CG33703, a protein whose functions remains uncharacterized despite having 75 paralogs listed in *Drosophila* genome (*Öztürk-Çolak et al., 2024*). Together with three of these paralogs (CG33783, CG33647, and CG33644), CG33703 was predicted to form dimer with Piwi (ranking confidences 0.82) (*Table 4*, *Figure 5—figure supplement 1C*). The domain of unknown function, DUF1091 (*Letunic et al., 2021*), shared by these paralogs was predicted to associate with the PIWI-domain (*Figure 5Biii*). Although these proteins are generally not expressed under the normal conditions (*Öztürk-Çolak et al., 2024*), their potential to bind Piwi suggests a regulatory role in the abnormal or stress conditions where CG33703 or its paralogs are expressed. In addition, we investigated two oogenesis-related proteins, Twinfilin (Twf, ranking confidence 0.64, *Figure 5Biv*) and Brainiac (Brn, ranking confidence 0.63, *Figure 5Bv*), for their binding with Piwi through co-immunoprecipitation (*Figure 5C*, *Supplementary file 7*). While no binding was observed with Twf, significant binding was detected with Brn, which is involved in dorsal-ventral polarity determination in follicle cells (*Goode et al., 1996*).

This study identifies several potential protein interactions, but AlphaFold2 predictions require caution. Protein–protein interactions involve conformational changes and dependencies on ligands, ions, and cofactors, which AlphaFold2 does not consider, potentially reducing prediction accuracy. Notably, the presence of a high-scoring model in terms of structural complementarity does not guarantee that the interaction is biologically significant. The expression patterns of these candidate proteins within the organism are crucial for further validation of our findings. It is likely that these proteins interact when co-expressed in the same cellular context. Under typical growth conditions, these interactions might not occur; however, in stress or disease states where these proteins are upregulated, the likelihood of interaction increases, potentially implicating these interactions in the disruption of normal cellular functions and contributing to disease or tumorigenesis. Furthermore, in silico screening proves extremely valuable, especially when dealing with toxic bait proteins, as it allows us to narrow down the list of potential candidates and reduce the need for hazardous experimental procedures. Ultimately, establishing these potential interactions in vivo could significantly advance our understanding of protein functions under both normal and pathological conditions.

# Materials and methods

**Key resources table**

| Reagent type (species) or resource | Designation | Source or reference | Identifiers | Additional information |
|---|---|---|---|---|
| Gene (*Drosophila melanogaster*) | Vas | FlyBase | FBgn0283442 | |
| Gene (*D. melanogaster*) | Spn-E | FlyBase | FBgn0003483 | |
| Gene (*D .melanogaster*) | Tej | FlyBase | FBgn0033921 | |
| Gene (*D. melanogaster*) | Tapas | FlyBase | FBgn0027529 | |
| Gene (*D. melanogaster*) | Qin | FlyBase | FBgn0263974 | |
| Gene (*D. melanogaster*) | Kots | FlyBase | FBgn0038191 | |
| Gene (*D. melanogaster*) | Krimp | FlyBase | FBgn0034098 | |
| Gene (*D. melanogaster*) | Squ | FlyBase | FBgn0267347 | |
| Gene (*D. melanogaster*) | Mael | FlyBase | FBgn0016034 | |
| Gene (*D. melanogaster*) | Aub | FlyBase | FBgn0000146 | |
| Gene (*D. melanogaster*) | AGO3 | FlyBase | FBgn0250816 | |
| Gene (*D. melanogaster*) | Papi | FlyBase | FBgn0031401 | |

*Continued on next page*

*Continued*

| Reagent type (species) or resource | Designation | Source or reference | Identifiers | Additional information |
|---|---|---|---|---|
| Gene (*D. melanogaster*) | Vret | FlyBase | FBgn0263143 | |
| Gene (*D. melanogaster*) | Bel | FlyBase | FBgn0263231 | |
| Gene (*D. melanogaster*) | Zuc | FlyBase | FBgn0261266 | |
| Gene (*D. melanogaster*) | Cup | FlyBase | FBgn0000392 | |
| Gene (*D. melanogaster*) | Tral | FlyBase | FBgn0041775 | |
| Gene (*D. melanogaster*) | Me31B | FlyBase | FBgn0004419 | |
| Gene (*D. melanogaster*) | Shu | FlyBase | FBgn0003401 | |
| Gene (*D. melanogaster*) | BoYb | FlyBase | FBgn0037205 | |
| Gene (*D. melanogaster*) | Piwi | FlyBase | FBgn0004872 | |
| Gene (*D. melanogaster*) | Mei-W68 | FlyBase | FBgn0002716 | |
| Gene (*D. melanogaster*) | CSN3 | FlyBase | FBgn0027055 | |
| Gene (*D. melanogaster*) | Pka-C1 | FlyBase | FBgn0000273 | |
| Gene (*D. melanogaster*) | Twf | FlyBase | FBgn0038206 | |
| Gene (*D. melanogaster*) | Brn | FlyBase | FBgn0000221 | |
| Gene (*D. melanogaster*) | Vps25 | FlyBase | FBgn0022027 | |
| Gene (*D. melanogaster*) | Nup44A | FlyBase | FBgn0033247 | |
| Gene (*D. melanogaster*) | Nclb | FlyBase | FBgn0263510 | |
| Gene (*D. melanogaster*) | Spn-D | FlyBase | FBgn0003482 | |
| Gene (*D. melanogaster*) | Jagn | FlyBase | FBgn0037374 | |
| Gene (*D. melanogaster*) | Rab7 | FlyBase | FBgn0015795 | |
| Gene (*D. melanogaster*) | Baf | FlyBase | FBgn0031977 | |
| Gene (*D. melanogaster*) | Mats | FlyBase | FBgn0038965 | |
| Gene (*D. melanogaster*) | Abo | FlyBase | FBgn0000018 | |
| Gene (*D. melanogaster*) | Rab11 | FlyBase | FBgn0015790 | |
| Gene (*D. melanogaster*) | Vls | FlyBase | FBgn0003978 | |
| Gene (*D. melanogaster*) | Hsc70-4 | FlyBase | FBgn0266599 | |
| Strain, strain background (*Escherichia coli*) | DH5α | Takara | Cat# 9057 | Competent cells |
| Genetic reagent (*D. melanogaster*) | w-; *squ*^HE47 cn bw/CyO; TM3 Sb/TM6 Tb | **Pane et al., 2007** | | |
| Genetic reagent (*D. melanogaster*) | w; *squ*^pp32/CyO; TM3 Sb/TM6 Tb | **Pane et al., 2007** | | |
| Cell line (*D. melanogaster*) | S2 | DRSC | FLYB:FBtc0000181; RRID:CVCL_Z992 | Cell line maintained in T. Kai lab |
| Antibody | Anti-Squ (rat polyclonal) | This study | | IF (1:5000) WB (1:1000) |
| Antibody | Anti-Spn-E (rat polyclonal) | **Lin et al., 2023** | | WB (1:500) |
| Antibody | Anti-Ago3 (rat polyclonal) | **Lin et al., 2023** | | WB (1:200) |
| Antibody | Anti-Aub (guinea pig polyclonal) | **Lim et al., 2022** | | WB (1:1000) |
| Antibody | Anti-Piwi (mouse monoclonal G-1) | Santa Cruz | Cat# sc-390946 | WB (1:1000) |

*Continued on next page*

*Continued*

| Reagent type (species) or resource | Designation | Source or reference | Identifiers | Additional information |
|---|---|---|---|---|
| Antibody | Anti-α-Tubulin (mouse monoclonal DM1A) | Santa Cruz | Cat# sc-32293; RRID:AB_628412 | WB (1:1000) |
| Antibody | Anti-guinea pig HRP-conjugated (rabbit polyclonal) | Dako | Cat # P0141; RRID:AB_628412 | WB (1:1000) |
| Antibody | Anti-rat HRP-conjugated (rabbit polyclonal) | Dako | Cat # P0450; RRID:AB_2630354 | WB (1:1000) |
| Antibody | Anti-mouse HRP-conjugated (goat polyclonal) | Bio-Rad | Cat # 1706516; RRID:AB_2921252 | WB (1:3000) |
| Antibody | Anti-rabbit HRP-conjugated (goat polyclonal) | Bio-Rad | Cat # 1706515; RRID:AB_11125142 | WB (1:3000) |
| Antibody | Anti-DDDDK-tag HRP-conjugated (mouse monoclonal) | MBL | Cat# M185-7; RRID:AB_2687989 | WB (1:1000) |
| Antibody | Anti-Myc-tag HRP-conjugated (mouse monoclonal) | MBL | Cat# M192-7; RRID:AB_3678890 | WB (1:1000) |
| Recombinant DNA reagent | Spn-E (plasmid) | *Lin et al., 2023* | | Myc-tag mK2-tag |
| Recombinant DNA reagent | Aub (plasmid) | *Patil and Kai, 2010* | | Myc-tag |
| Recombinant DNA reagent | BoYb (plasmid) | This study | | Myc-tag Flag-tag |
| Recombinant DNA reagent | Me31B (plasmid) | This study | | Myc-tag |
| Recombinant DNA reagent | Vret (plasmid) | This study | | Flag-tag |
| Recombinant DNA reagent | Shu (plasmid) | This study | | Flag-tag |
| Recombinant DNA reagent | Squ$^{WT}$ (plasmid) | This study | | Flag-tag GFP tag |
| Recombinant DNA reagent | Squ$^{4A}$ (plasmid) | This study | | Flag-tag GFP tag |
| Recombinant DNA reagent | Squ$^{E107A}$ (plasmid) | This study | | Flag-tag GFP tag |
| Recombinant DNA reagent | Squ$^{E109A}$ (plasmid) | This study | | Flag-tag GFP tag |
| Recombinant DNA reagent | Squ$^{R115A}$ (plasmid) | This study | | Flag-tag GFP tag |
| Recombinant DNA reagent | Squ$^{K163A}$ (plasmid) | This study | | Flag-tag GFP tag |
| Recombinant DNA reagent | Tej (plasmid) | *Patil and Kai, 2010* | | Flag-tag |
| Recombinant DNA reagent | Vas (plasmid) | *Patil and Kai, 2010* | | Flag-tag |
| Recombinant DNA reagent | Mei-W68 (plasmid) | This study | | Myc-tag |
| Recombinant DNA reagent | CSN3 (plasmid) | This study | | Myc-tag |
| Recombinant DNA reagent | Pka-C1 (plasmid) | This study | | Myc-tag |
| Recombinant DNA reagent | Vps25 (plasmid) | This study | | Myc-tag |
| Recombinant DNA reagent | Nup44A (plasmid) | This study | | Myc-tag |
| Recombinant DNA reagent | Nclb (plasmid) | This study | | Myc-tag |
| Recombinant DNA reagent | Spn-D (plasmid) | This study | | Myc-tag |
| Recombinant DNA reagent | Jagn (plasmid) | This study | | Myc-tag |
| Recombinant DNA reagent | Rab7 (plasmid) | This study | | Myc-tag |

*Continued on next page*

*Continued*

| Reagent type (species) or resource | Designation | Source or reference | Identifiers | Additional information |
|---|---|---|---|---|
| Recombinant DNA reagent | Baf (plasmid) | This study | | Myc-tag |
| Recombinant DNA reagent | Mats (plasmid) | This study | | Myc-tag |
| Recombinant DNA reagent | Abo (plasmid) | This study | | Myc-tag |
| Recombinant DNA reagent | Rab11 (plasmid) | This study | | Myc-tag |
| Recombinant DNA reagent | Vls (plasmid) | This study | | Myc-tag |
| Recombinant DNA reagent | Hsc70-4 (plasmid) | This study | | Myc-tag |
| Recombinant DNA reagent | Piwi (plasmid) | This study | | Flag-tag |
| Recombinant DNA reagent | Twf (plasmid) | This study | | Myc-tag |
| Recombinant DNA reagent | Brn (plasmid) | This study | | Myc-tag |
| Commercial assay or kit | anti-FLAG magnetic beads | MBL | Cat# M185-11R | |
| Commercial assay or kit | anti-Myc magnetic beads | Thermo Fisher | Cat# 88842 | |
| Commercial assay or kit | Dynabeads protein A | Thermo Fisher | Cat# 10001D | |
| Commercial assay or kit | Dynabeads protein G | Thermo Fisher | Cat# 10003D | |
| Chemical compound, drug | Hilymax | Dojindo | Cat# 342-91103 | Transfection in S2 |
| Chemical compound, drug | Signal Enhancer HIKARI | Nacalai Tesque | Cat# 02270-81 | Western blotting |
| Chemical compound, drug | Chemi-Lumi One reagent kit | Nacalai Tesque | Cat# 07880-54 | Western blotting |
| Chemical compound, drug | Fluoro-Keeper Antifade Reagent | Nacalai Tesque | Cat# 12593-64 | |
| Software, algorithm | AlphaFold v2.2 | Developed by DeepMind | RRID:SCR_025454 | Installed in SQUID (Osaka University) |
| Software, algorithm | ImageJ | *Schneider et al., 2012* | | |

## Antibodies

The anti-Squ antibody was generated as follows. His-tagged full-length Squ was expressed in *Escherichia coli* BL21(DE3) strain, with the plasmid that subcloned the *squ* coding region into pDEST17 vector (Thermo Fisher Scientific). His-Squ was solubilized with 6 M urea in PBS, purified using Nickel Sepharose beads (GE healthcare) following the manufacturer's protocol, and subsequently used for immunization in rats. The antibodies used for western blotting analysis were rat anti-Spn-E[17] (1:500), rat anti-Ago3[17] (1:200), guinea pig anti-Aub (*Lim et al., 2022*) (1:1000), mouse monoclonal anti-Piwi (G-1, sc-390946, Santa Cruz Biotechnology, USA), and mouse monoclonal anti-α-Tubulin (DM1A, sc-32293, Santa Cruz Biotechnology). The secondary antibodies used in this study were HRP-conjugated goat anti-guinea pig (Dako, Cat# P0141), HRP-conjugated goat anti-rat (Dako, Cat# P0450), HRP-conjugated goat anti-mouse (Bio-Rad, Cat# 1706516), and HRP-conjugated goat anti-rabbit (Bio-Rad, Cat# 1706515). HRP-conjugated anti-DDDDK-tag antibody (MBL, Cat# M185-7) and HRP-conjugated anti-Myc-tag antibody (MBL, Cat# M192-7) were used to detect FLAG-tagged and Myc-tagged proteins, respectively.

## AlphaFold2 prediction for the direct interacting protein pairs

Amino acid sequences for *Drosophila* proteins were obtained from FlyBase (*Öztürk-Çolak et al., 2024*). For proteins annotated with multiple isoforms, only the longest isoform was selected. Proteins exceeding 2000 residues were excluded due to computational limitations. AlphaFold v2.2 program was installed in the Supercomputer for Quest to Unsolved Interdisciplinary Datascience (SQUID) at the Cyber Media Center in Osaka University. All necessary protein sequence databases for AlphaFold2 were stored on an SSD device connected to the SQUID system.

The AlphaFold2 prediction process was divided into two steps: generation of the multiple sequence alignment (MSA) and the prediction of the 3D structure. The MSAs were computed on SQUID's CPU

node and stored for reuse. The calculation of the MSA took on average 2–4 h per protein, with the more homologs of the protein in query, the longer it took. For dimer structure prediction, two MSAs corresponding to the dimer pair were placed in the directory of msas/A and msas/B. The calculations were performed on the GPU node with the options of -t 2022-05-14 -m multimer -l 1 -p true. Alpha-Fold2 generates five structural models for each prediction. To speed up the prediction, five computations were assigned to five GPU units, even though the original AlphaFold2 program computes five models one at a time. Prediction of dimer structure took approximately 1–2 h per pair on average, depending on protein size. Each user can compute 100–200 pairs of calculations per day, but since the supercomputer is shared, job availability varies with overall demand.

The prediction confidence score (ranking confidence) was provided for each model, and among five models, the highest ranking confidence was used as the prediction score for the corresponding dimer structure. PAE plots for dimer structures were drawn by extracting the data form pkl files generated by AlphaFold2. The list of protein pairs scoring above 0.6 and the corresponding PAE plots and PDB structures is available on GitHub (https://dme-research.github.io/AF2_2/).

## AlphaFold3 prediction for the structure of the trimer complex

The structure of Spn-E_Squ_Tej complexed with RNA, 5'-CUGACUACCGAAGUACUACG-3' was predicted by the AlphaFold3 prediction server (https://alphafoldserver.com/) (**Abramson et al., 2024**). The trimer structures of Spn-E_Squ_Tej, Vas_Tej_Spn-E, BoYb_Vret_Shu, and Me31B_Cup_Tral were also predicted by AlphaFold3.

## Analysis of protein 3D structure

The protein 3D structure was visualized using ChimeraX software (**Pettersen et al., 2021**). The SpnE_Squ dimer interface was analyzed with the 'Protein interfaces, surfaces and assemblies' service (PISA) at the European Bioinformatics Institute (http://www.ebi.ac.uk/pdbe/prot_int/pistart.html; **Krissinel and Henrick, 2007**).

## Fly stocks

All stocks were maintained at 25°C with standard methods. Mutant alleles of *squ* (*squ^pp32* and *squ^HE47*) were used in this study (**Pane et al., 2007**). The mK2-tagged Spn-E-mK2 knock-in fly was previously generated (**Lin et al., 2023**). *y w* strain served as the control.

## Western blotting

Ovaries were homogenized in the ice-cold PBS and denatured in the presence of SDS sample buffer at 95°C for 5 min. The samples were then subjected to SDS-PAGE and transferred to ClearTrans SP PVDF membrane (Wako). The primary and secondary antibodies described above were diluted in the Signal Enhancer reagent HIKARI (Nacalai Tesque). Chemiluminescence was induced by the Chemi-Lumi One reagent kit (Nacalai Tesque) and detected with ChemiDoc Touch (Bio-Rad). The bands were quantified using ImageJ (**Schneider et al., 2012**) or Image Lab software (Bio-Rad).

## Co-immunoprecipitation in S2 cells

The *Drosophila Schneider* S2 cell line (S2-DRSC), derived from *D. melanogaster* embryos, was obtained from the Drosophila Genomics Resource Centre (DGRC) and is not listed among commonly misidentified cell lines. The S2 cells were cultured at 28°C in Schneider's medium supplemented with 10% (v/v) fetal bovine serum and antibiotics (penicillin and streptomycin). Mycoplasma contamination was not detected using the VenorGeM Classic Mycoplasma Detection Kit (Minerva Biolabs). Protein coding regions were cloned into pENTR vector (Thermo Fisher Scientific) and then transferred into pAFW or pAMW destination vectors. S2 cells (0.2–2 × 10⁶ cells/ml) were seeded in 12-well plates overnight and transfected using Hilymax (Dojindo Molecular Technologies, Japan). After 36–48 h, S2 cells were resuspended in 360 µl of ice-cold PBS containing 0.02% Triton-X100 and 1× protease inhibitor cocktail (Roche), and sonicated (0.5 s, five times). The resulted lysate was clarified by spinning at 15,000 × $g$ for 15 min at 4°C. 300 µl of supernatant was incubated with 6 µl of prewashed anti-FLAG magnetic beads (MBL) or anti-Myc magnetic beads (Thermo Fisher Scientific) for 1.5 h at 4°C with gentle rotation. After incubation, the beads were washed three times with 800 µl of ice-cold PBS with

0.02% Triton-X100, denatured in SDS sample buffer and subjected to SDS-PAGE and western blot. 1% of the total lysates were loaded as input samples.

## Co-localization assay in S2 cells

Construction of GFP-tagged or mKate2-tagged proteins and transfection were conducted as described in the previous section. After 48 h of transfection, the cells were placed onto the concanavalin A-coated coverslips for 20 min, fixed with PBS containing 4% (w/v) paraformaldehyde for 15 min at room temperature, permeabilized with PBX (PBS containing 0.2% [v/v] TritonX-100) for 10 min twice, stained with DAPI (1:1000) and mounted with Fluoro-Keeper Antifade Reagent (Nacalai Tesque). Images were taken by ZEISS LSM 900 with Airy Scan 2 using ×63 oil NA 6.0 objectives and processed using ZEISS ZEN 3.0 and ImageJ (*Schneider et al., 2012*).

## Crosslinking immunoprecipitation (CL-IP)

As previously described (*Lin et al., 2023*), 100 ovaries from *y w* flies were dissected in ice-cold PBS and fixed in PBS containing 0.1% (w/v) paraformaldehyde for 20 min on ice, quenched in 125 mM glycine for 20 min, and then homogenized in CL-IP lysis buffer. The lysate was incubated at 4°C for 20 min and then sonicated. After centrifugation at maximum speed for 10 min at 4°C, the supernatant was collected and diluted with an equal volume of CL-IP wash buffer. 10 μl of pre-washed Dynabeads Protein G/A mixture (1:1) (Invitrogen) was added for pre-clearance at 4°C for 1 h. Anti-Squ antibody was added to the cleared supernatant with 1:500 dilution and incubated at 4°C overnight. The 20 μl of pre-washed Dynabeads Protein G/A 1:1 mixture beads (Invitrogen) were added for binding and incubated at 4°C for 3 h. After washed with CL-IP wash buffer for three times, beads were collected and 50 μl of CL-IP wash buffer containing SDS sample buffer was added. The beads were boiled at 95°C for 5 min and subjected for SDS-PAGE and western blotting analysis.

## Immunostaining of ovaries

As previously described (*Lin et al., 2023*; *Lim et al., 2022*), ovaries were dissected, fixed, permeabilized with PBX and immunostained. The primary and the secondary antibodies were anti-Squ antibody (in this study, 1:500) and Alexa Fluor 488-conjugated anti-rat IgG (Thermo Fisher Scientific, 1:200). Images were taken by ZEISS LSM 900 with Airy Scan 2 using ×63 oil NA 1.4 objectives and processed by ZEISS ZEN 3.0 and ImageJ (*Schneider et al., 2012*).

# Acknowledgements

The prediction by AlphaFold2 was achieved through the use of large-scale computer systems, Supercomputer for Quest to Unsolved Interdisciplinary Datascience (SQUID) at the D3 Center, Osaka University through the Research Proposal-based Use, Large-Scale High-Performance Computing Projects to SK (D3 center, Osaka University) and the High Performance Computing Infrastructure (HPCI) System Research Project (Project ID: hp240099 to SK). We thank Dr Trudi Schüpbach (Princeton University) for generous gifts of squ mutant flies. We also thank the FBS Core Facility in Osaka University for providing access to the LSM 900 and ChemiDoc Touch. We appreciate the insightful discussion and suggestions from all the members of TK's laboratory. This work was supported by TAKEDA Bioscience Research Grant (J191503009 to TK); Grant-in-Aid for Transformative Research Areas (A) (21H05275 to TK); and Osaka University Institute for Datability Science 'Transdisciplinary Research Project' (Na22990007 to SK, SD and TK).

# Additional information

## Competing interests

Kenta Yamaguchi, Ryuuya Kawasaki: Employee of NEC Solution Innovators, Ltd. The other authors declare that no competing interests exist.

## Funding

| Funder | Grant reference number | Author |
|---|---|---|
| Takeda Science Foundation | J191503009 | Toshie Kai |
| Ministry of Education, Culture, Sports, Science and Technology | 21H05275 | Toshie Kai |
| Osaka University | Na22990007 | Shinichi Kawaguchi Susumu Date Toshie Kai |

The funders had no role in study design, data collection and interpretation, or the decision to submit the work for publication.

## Author contributions

Shinichi Kawaguchi, Conceptualization, Resources, Data curation, Software, Formal analysis, Supervision, Funding acquisition, Validation, Investigation, Visualization, Methodology, Writing – original draft, Project administration, Writing – review and editing; Xin Xu, Formal analysis, Investigation, Visualization, Methodology, Writing – original draft, Writing – review and editing; Takashi Soga, Software, Methodology; Kenta Yamaguchi, Ryuuya Kawasaki, Ryota Shimouchi, Software; Susumu Date, Software, Funding acquisition, Methodology; Toshie Kai, Resources, Supervision, Funding acquisition, Writing – original draft, Project administration, Writing – review and editing

## Author ORCIDs

Shinichi Kawaguchi (ORCID) https://orcid.org/0000-0002-7832-1918
Toshie Kai (ORCID) https://orcid.org/0000-0001-8675-8469

Reviewer #1 (Public review): https://doi.org/10.7554/eLife.101967.3.sa1
Reviewer #2 (Public review): https://doi.org/10.7554/eLife.101967.3.sa2
Author response https://doi.org/10.7554/eLife.101967.3.sa3

# Additional files

## Supplementary files

Supplementary file 1. The prediction confidence scores (ranking confidences) for the pairwise dimer predictions by AlphaFold2 as shown in *Figure 1A*.

Supplementary file 2. The experimentally determined 3D structure models for the nuage-localizing/piRNA-related proteins.

Supplementary file 3. The salt-bridges and H-bonds found in the predicted interface between Spn-E and Squ dimer.

Supplementary file 4. The hydrophobic residues found in the predicted interface between Spn-E and Squ proteins.

Supplementary file 5. The AlphaFold2 screening for the interacting pairs of Squ, Tej, Vas against 430 oogenesis-related proteins.

Supplementary file 6. The computed binding affinity of the protein–protein complex on the basis of the three-dimensional structure predicted by AlphaFold2.

Supplementary file 7. The AlphaFold2 screen for Piwi interacting pairs against 12,427 *Drosophila* proteins.

Supplementary file 8. The list of oligo DNA primers used in this study.

MDAR checklist

## Data availability

PDB files and PAE plots for the protein dimers whose ranking confidences were more than 0.6 were deposited and available at GitHub (https://dme-research.github.io/AF2_2/). The data supporting the

findings of this study are available within the article and its supplementary and source data files. The anti-Squ antibody is available from the corresponding authors upon reasonable request.

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
