## [Editor Report · eLife Assessment]

This **useful** study employs AlphaFold2 to predict interactions among 20 nuage proteins, identifying 5 novel interaction candidates, 3 of which are validated experimentally through co-immunoprecipitation. Expanding the analysis to 430 oogenesis-related proteins and screening ~12,000 *Drosophila* proteins for interactions with Piwi, the study identifies 164 potential binding partners, demonstrating how computational predictions can streamline experimental validation. This study provides a **solid** basis for further investigations into eukaryotic protein interaction networks.

---

## [Referee Report · Reviewer #1 (Public review)]

Summary:

The study investigates protein-protein interactions (PPIs) within the nuage, a germline-specific organelle essential for piRNA biogenesis in *Drosophila melanogaster*, using AlphaFold2 to predict interactions among 20 nuage-localizing proteins. The authors identify five novel interaction candidates and experimentally validate three of them, including Spindle-E and Squash, through co-immunoprecipitation assays. They confirm the functional significance of these interactions by disrupting salt bridges at the Spn-E_Squ interface. The study further expands its scope to analyze approximately 430 oogenesis-related proteins, validating three additional interaction pairs. A comprehensive screen of around 12,000 *Drosophila* proteins for interactions with the key piRNA pathway player, Piwi, identifies 164 potential binding partners. Overall, the research demonstrates that in silico approaches using AlphaFold2 can link bioinformatics predictions with experimental validation, streamlining the identification of novel protein interactions and reducing the reliance on extensive experimental efforts.

---

## [Referee Report · Reviewer #2 (Public review)]

Summary:

In this paper, the authors use AlphaFold2 to identify potential binding partners of nuage localizing proteins.

Strengths:

The main strength of the paper is that the authors experimentally verify a subset of the predicted interactions.

Many studies have been performed to predict protein-protein interactions in various subsets of proteins. The interesting story here is that the authors (i) focus on an organelle that contains quite some intrinsically disordered proteins and (ii) experimentally verify some (but not all) predictions.

Weaknesses:

Identification of pairwise interactions is only a first step towards understanding complex interactions. It is pretty clear from the predictions that some (but certainly not all) of the pairs could be used to build larger complexes. This is Done only for some cases and could be extended to the entire network.

---

## [Author Response]

The following is the authors’ response to the original reviews.

**Public Reviews:**

**Reviewer #1 (Public review):**
Summary:The study investigates protein-protein interactions (PPIs) within the nuage, a germline-specific organelle essential for piRNA biogenesis in *Drosophila melanogaster*, using AlphaFold2 to predict interactions among 20 nuage-localizing proteins. The authors identify five novel interaction candidates and experimentally validate three of them, including Spindle-E and Squash, through co-immunoprecipitation assays. They confirm the functional significance of these interactions by disrupting salt bridges at the Spn-E_Squ interface. The study further expands its scope to analyze approximately 430 oogenesis-related proteins, validating three additional interaction pairs. A comprehensive screen of around 12,000 Drosophila proteins for interactions with the key piRNA pathway player, Piwi, identifies 164 potential binding partners. Overall, the research demonstrates that in silico approaches using AlphaFold2 can link bioinformatics predictions with experimental validation, streamlining the identification of novel protein interactions and reducing the reliance on extensive experimental efforts. The manuscript is commendably clear and easy to follow; however, areas for improvement should be addressed to enhance its clarity and rigor.Major Concerns:(1) While AlphaFold2 was developed and trained primarily for predicting protein structures and their interactions, applying it to predict protein-protein interactions is an extrapolation of its intended use. This introduces several important considerations and risks. First, it assumes that AlphaFold's accuracy in structure prediction extends to interactions, despite not being explicitly trained for this task. Additionally, the assumption that high-scoring models with structural complementarity imply biologically relevant interactions is not always valid. Experimental validation is essential to address these uncertainties, as over-reliance on computational predictions without such validation can lead to false positives and inaccurate conclusions. The authors should expand on the assumptions, limitations, and risks associated with using AlphaFold2 for predicting protein-protein interactions.

We appreciate the reviewer's point. The prediction of protein-protein interactions using AlphaFold2 relies on the number of conserved homologous sequences and previous conformational data(8) (Jumper, J. et al. Highly accurate protein structure prediction with AlphaFold. Nature 596, 583–589 (2021)). We added sentences explaining the limitations and risks of the AlphaFold2 prediction method in Introduction and the end of Result and Discussion of the revised manuscript, respectively.

Page 5, Line 67;

“AlphaFold2 requires sequence homology information to predict protein-protein interactions and the complex structure model. The reliability of these predictions is basically dependent on the strength of co-evolutionary signals(9).”

Page 6, Line 84;

“AlphaFold2 was initially trained to predict the structure of individual proteins(8). Its application to complex prediction is an extrapolative use beyond its original intended scope, and its accuracy remains unverified. Even high-confidence predictions may not correspond to actual interactions, necessitating experimental validation to confirm whether predicted protein dimers truly bind.”

Page 21, Line 361;

“This study identifies several potential protein interactions, but AlphaFold2 predictions require caution. Protein-protein interactions involve conformational changes and dependencies on ligands, ions, and cofactors, which AlphaFold2 does not consider, potentially reducing prediction accuracy. Notably, the presence of a high-scoring model in terms of structural complementarity does not guarantee that the interaction is biologically significant.”

(2) The authors experimentally validated three interactions, out of five predicted interactions, using co-immunoprecipitation (co-IP). They attributed the lack of validation for the other two predictions to the limitations of the co-IP method. However, further clarification on the potential limitations of the co-immunoprecipitation behind the negative results would strengthen the conclusions. While co-IP is a widely used technique, it may not detect weak or transient interactions, which could explain the failure to validate some predictions. Suggesting alternative validation methods such as FRET or mass spectrometry could further substantiate the results. On the other hand, AlphaFold2 predictions are not infallible and may generate false positives, particularly when dealing with structurally plausible but biologically irrelevant interactions. By acknowledging both the potential limitations of co-IP and the possibility of false positives from AlphaFold2, the authors can provide a more balanced interpretation of their findings.

We appreciate the reviewer's point of view. We have used the co-IP method to detect interactions in this study. However, as the reviewer pointed out, it is likely that weak and transient interactions may not be detected. We added a note on the detection limits of the co-IP method and the possibility that AlphaFold2 method produces false positives in the revised manuscript.

Page 12, Line 197;

“While co-immunoprecipitation is a widely used method, it may not always detect weak or transient interactions. Other validation methods, such as FRET or co-localization assay in culture cells, could offer further insights to support the results. It is also important to note that AlphaFold2's predictions are not definitive and may lead to false positives, particularly when analyzing a large number of interactions.”

(3) In line 143, the authors state that "This approach identified 13 pairs; seven of these were already known to form complexes, confirming the effectiveness of AlphaFold2 in predicting complex formations (Table 2). The highest pcScore pair was the Zuc homodimer, possibly because AlphaFold2 had learned from Zuc homodimer's crystal structure registered in the database." While the authors mentioned the presence of the Zuc homodimer's crystal structure, they do not provide a systematic bioinformatics analysis to evaluate pairwise sequence identity or check for the presence of existing structures for all the proteins or protein pairs (or their homologs) in databases such as the Protein Data Bank (PDB) or Swiss-Model. Conducting such an analysis is critical, as it significantly impacts the novelty and reliability of AlphaFold2 predictions. For instance, high sequence identity between the query proteins could lead to high-scoring models for biologically irrelevant interactions. Including this information would strengthen the conclusions regarding the accuracy and utility of the predictions.

We appreciate the reviewer's critical point. The AlphaFold2 method generates a high confidence score when the 3D structure of the protein of interest, or of proteins with very similar sequences, is solved. We investigated whether the proteins used in this study are included in the 3D structure database (PDB) and added the information as a supplemental table S2. The following sentences were added to explain the structural references that AlphaFold2 has learned in the revised manuscript.

Page 9, Line 150;

The structures of the 20 proteins used in this study have been analyzed to varying extents in previous studies (Supplementary Table S2). A complex of Vas and the Lotus domain of Osk has been reported(20), and based on this complex structure, the interaction between Vas and Tej Lotus domain was predicted with a high score. Although the conformational analyses of the RNA helicase domain and the eTud domain have been reported previously, many of those cover only a subset of the regions and unlikely to affect our predictions in this study.

The predicted 3D structures and the Predicted Aligned Error (PAE) plots for the 12 pairs, are shown in Fig. 1C.

(4) While the manuscript successfully identifies novel protein interactions, the broader biological significance of these interactions remains underexplored. The manuscript could benefit from elaborating on how these findings may contribute to understanding the piRNA pathway and its implications on germline development, transposon repression, and oogenesis.

We added to the revise manuscript the potential biological significance of the novel protein-protein interactions presented in this manuscript as follows;

Page 16, Line 268;

“In this study, three novel protein-protein interactions were predicted and experimentally confirmed. AlphaFold2 also predicted the 3D structure of these complexes, providing insight into the important regions involved in complex formation. These predictions will provide fundamental information to elucidate nuage assembly. Nuage is thought to form by liquid-phase separation; however, direct protein-protein interactions likely occur within protein-dense nuage, facilitating RNA processing. Although the precise roles of individual interactions require further study, characterization of protein-protein interactions within nuage will help clarify the mechanism of piRNA production.”

**Reviewer #1 (Recommendations for the authors):**
Minor Concerns:(1) In the Materials and Methods section, the authors thoroughly describe the computational infrastructure (SQUID at Osaka University) and the use of AlphaFold2. However, it would greatly benefit the readers to include a detailed breakdown of the computational cost. Understanding the computational cost (in terms of time, CPU/GPU hours, or other relevant metrics) for predicting 3D structures, especially for 400 protein pairs, would provide valuable insight into the efficiency and scalability of the approach. This would enhance the practical relevance of the methodology section and offer a better understanding of the resources required, beyond just the infrastructure description.

Thank you for your valuable suggestion. The following descriptions were added in the revised manuscript.

Page 24, Line 403;

“The calculation of the MSA took on average 2-4 hours per protein, with the more homologs of the protein in query, the longer it took.”

Page 24, Line 409;

“Prediction of dimer structure took approximately 1-2 hours per pair on average, depending on protein size. Each user can compute 100~200 pairs of calculations per day, but since the supercomputer is shared, job availability varies with overall demand.”

(2) The manuscript will benefit from a review for grammatical accuracy and clarity, especially in complex explanations. For example, in Line 160: "The predicted dimer structures of Me31B_Tral and Cup_Me31B showed the score of 0.74 and 0.68, respectively (Table 2)." could be revised to "The predicted dimer structures of Me31B_Tral and Cup_Me31B showed scores of 0.74 and 0.68, respectively.

Thank you very much for pointing it out. Correction has been made to the text pointed out (Page 10, Line 170).

(3) For alphafold3 webserver, please use (https://alphafoldserver.com/) instead of (https://golgi.sandbox.google.com/about).

Thank you very much for pointing it out. The URL has been changed in the revised manuscript (Page 25, Line 422).

**Reviewer #2 (Public review):**
Summary:In this paper, the authors use AlphaFold2 to identify potential binding partners of nuage localizing proteins.Strengths:The main strength of the paper is that the authors experimentally verify a subset of the predicted interactions.Many studies have been performed to predict protein-protein interactions in various subsets of proteins. The interesting story here is that the authors (i) focus on an organelle that contains quite some intrinsically disordered proteins and (ii) experimentally verify some (but not all) predictions.Weaknesses:Identification of pairwise interactions is only a first step towards understanding complex interactions. It is pretty clear from the predictions that some (but certainly not all) of the pairs could be used to build larger complexes. AlphaFold easily handles proteins up to 4-5000 residues, so this should be possible. I suggest that the authors do this to provide more biological insights.

We thank the reviewer for his kind suggestions. In this study, protein dimers were screened on the assumption that the two proteins bind 1:1; in some cases, multiple binding partners were predicted for a single protein. For example, Spn-E was predicted to bind Tej and Squ, respectively. Therefore, for Spn-E_Squ_Tej, we used the latest AlphaFold3 to predict the trimeric structure, which has already been described in the first manuscript. In addition, as suggested by the reviewer, other possible trimer results were also added in the revised manuscript as follows;

Page 15, Line 249;

“In addition to the Spn-E_Squ_Tej complex, 1:1 dimer prediction described above further suggested potential trimers (Fig. 1; Supplemental Fig. S4). For example, Tej protein is predicted to bind both Vas and Spn-E, and AlfaFold3 indeed further predicted a Vas_Tej_Spn-E trimer, where Tej’s Lotus and eTud domains interact with Vas and Spn-E, respectively. However, Lin et al. reported that Tej binds exclusively either with Vas or Spn-E, but not simultaneously(17), in *Drosophila* ovary, suggesting that the predicted trimers may be weak or transient. Similarly, the BoYb_Vret_Shu and the Me31B_Cup_Tral trimers remain hypothetical and require experimental verification (Supplemental Fig. S4).”

Another weakness is the use of a non-standard name for "ranking confidence" - the author calls it the pcScore - while the name used in AlphaFold (and many other publications) is ranking confidence.

“pcScore” has been changed to “ranking confidence”

**Reviewer #2 (Recommendations for the authors):**
(1) The pcScore is actually what is called RankingConfidence. Also, many other measures have been developed by other groups (based on PAE for instance) - these could be compared.

Thank you for your valuable suggestions. While other indicators are being developed, we have computed the affinity of the complex based on the predicted three-dimensional structure by using PRODIGY web server. The description was added in the revised manuscript as follows;

Page 18, Line 300;

“The ranking confidence score reflects the reliability of AlphaFold2's predicted structure but does not always ensure accuracy. Therefore, we assessed complex affinity based on the predicted three-dimensional structures (Supplemental Table S6). Most dimers with high ranking confidence scores exhibited low Kd values indicative of high affinity, while some showed high Kd values indicating weak interactions (Supplemental Table S6). For example, the Baf_Vas complex had a high AlphaFold2 ranking confidence score (0.85) but a relatively high Kd value (1.1E-4 M), indicating low affinity. Consistently, Baf_Vas binding was not detected in Co-IP experiments (Fig. S5C). Although accurate Kd prediction may be limited due to insufficient structural optimization, it could serve as a valuable secondary screening tool following AlphaFold2 predictions.”

(2) A statistical estimate of FDR for binding to the PIWI protein needs to be estimated. It is possible that 1.6% of random proteins (from another species for instance) also obtain ranking confidence over 0.6, i.e. how trustful are the predictions?

Thank you for the insightful comments. Unfortunately, it is difficult to infer the FDR from the value of ranking confidence. Presumably, the accuracy will vary depending on the target protein, since the number of homologs and known conformational information will differ. In the case of Piwi, the FDR is expected to be relatively low since the conformation of the protein on its own has been experimentally determined. However, even for Piwi complexes with high values of ranking confidence, the estimated affinity varied from high to low (Supplemental Table S6). Therefore, it may be useful to conduct further secondary evaluation for AlphaFold2 predictions with high ranking confidence.

(3) Identification of pairwise interactions is only a first step towards understanding complex interactions. It is pretty clear from the predictions that some (but certainly not all) of the pairs could be used to build larger complexes. AlphaFold easily handles proteins up to 4-5000 residues, so this should be possible. I suggest that the authors do this to provide more biological insights.

Already mentioned above.

(4) The comparisons of ranking confidence vs ipTM/pTM are less interesting (by definition ranking confidence is virtually identical to ipTM).

Thank you for the thoughtful comment. As the reviewer pointed out, there is not much difference between ranking confidence and ipTM shown in Fig. 1A. A high value of pTM (firmly folding) tends to increase ranking confidence, while a low value of pTM (many disorder regions) tends to decrease ranking confidence. Therefore, it may be useful to change the threshold for confidence for each protein pair.